# OVER-PARAMETERIZATION IMPROVES GENERALIZATION IN THE XOR DETECTION PROBLEM

## ABSTRACT

Empirical evidence suggests that neural networks with ReLU activations generalize better with over-parameterization. However, there is currently no theoretical analysis that explains this observation. In this work, we study a simplified learning task with over-parameterized convolutional networks that empirically exhibits the same qualitative phenomenon. For this setting, we provide a theoretical analysis of the optimization and generalization performance of gradient descent. Specifically, we prove data-dependent sample complexity bounds which show that over-parameterization improves the generalization performance of gradient descent.

## 1 INTRODUCTION

Most successful deep learning models use a number of parameters that is larger than the number of parameters that are needed to get zero-training error. This is typically referred to as *over-parameterization*. Indeed, it can be argued that over-parameterization is one of the key techniques that has led to the remarkable success of neural networks. However, there is still no theoretical account for its effectiveness.

One very intriguing observation in this context is that over-parameterized networks with ReLU activations, which are trained with gradient based methods, often exhibit better generalization error than smaller networks (Neyshabur et al., 2014; 2018; Novak et al., 2018). This somewhat counter-intuitive observation suggests that first-order methods which are trained on over-parameterized networks have an *inductive bias* towards solutions with better generalization performance. Understanding this inductive bias is a necessary step towards a full understanding of neural networks in practice.

Providing theoretical guarantees for this phenomenon is extremely challenging due to two main reasons. First, to show a generalization gap, one needs to prove that large networks have better sample complexity than smaller ones. However, current generalization bounds that are based on complexity measures do not offer such guarantees. Second, analyzing the dynamics of first-order methods on networks with ReLU activations is a major challenge. Indeed, there do not exist optimization guarantees even for simple learning tasks such as the classic XOR problem in two dimensions. [1]

To advance this issue, we focus on a particular learning setting that captures key properties of the over-parameterization phenomenon. We consider a high-dimensional extension of the XOR problem, which we refer to as the "XOR Detection problem (XORD)". The XORD is a pattern recognition task where the goal is to learn a function which classifies binary vectors according to whether they contain a two-dimensional binary XOR pattern (i.e., $(1,1)$ or $(-1,-1)$). This problem contains the classic XOR problem as a special case when the vectors are two dimensional. We consider learning this function with gradient descent trained on an over-parameterized convolutional neural network (i.e., with multiple channels) with ReLU activations and three layers: convolutional, max pooling and fully connected. As can be seen in Fig. 1, over-parameterization improves generalization in this problem as well. Therefore it serves as a good test-bed for understanding the role of over-parameterization.

---

[1] We are referring to the problem of learning the XOR function given four two-dimensional points with binary entries, using a moderate size one-hidden layer neural network (e.g., with 50 hidden neurons). Note that there are no optimization guarantees for this setting. Variants of XOR have been studied in Lisboa & Perantonis (1991); Sprinkhuizen-Kuyper & Boers (1998) but these works only analyzed the optimization landscape and did not provide guarantees for optimization methods. We provide guarantees for this problem in Sec. 9.

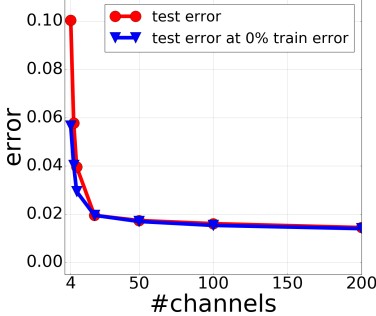

Figure 1: Over-parameterization improves generalization in the XORD problem. The network in Eq. 1 and Fig. 4 is trained on data from the XORD problem (see Sec. 3). The figure shows the test error obtained for different number of channels $k$. The blue curve shows test error when restricting to cases where training error was zero. It can be seen that increasing the number of channels improves the generalization performance. Experimental details are provided in Section 8.2.1.

.

In this work we provide an analysis of optimization and generalization of gradient descent for XORD. We show that for various input distributions, ranges of accuracy and confidence parameters, sufficiently over-parameterized networks have better sample complexity than a small network which can realize the ground truth classifier. To the best of our knowledge, this is the first example which shows that over-paramaterization can provably improve generalization for a neural network with ReLU activations.

Our analysis provides a clear distinction between the inductive bias of gradient descent for over-parameterized and small networks. It reveals that over-parameterized networks are biased towards global minima that detect more patterns in the data than global minima found by small networks. [2] Thus, even though both networks succeed in optimization, the larger one has better generalization performance. We provide experiments which show that the same phenomenon occurs in a more general setting with more patterns in the data and non-binary input. We further show that our analysis can predict the behavior of over-parameterized networks trained on MNIST and guide a compression scheme for over-parameterized networks with a mild loss in accuracy (Sec. 6).

## 2 RELATED WORK

In recent years there have been many works on theoretical aspects of deep learning. We will refer to those that are most relevant to this work. First, we note that we are not aware of any work that shows that generalization performance provably improves with over-parameterization. This distinguishes our work from all previous works.

Several works study convolutional networks with ReLU activations and their properties (Du et al., 2017a;b; Brutzkus & Globerson, 2017). All of these works consider convolutional networks with a single channel. Daniely (2017) and Li & Liang (2018) provide guarantees for SGD in general settings. However, their analysis holds for over-parameterized networks with an extremely large number of neurons that are not used in practice (e.g., the number of neurons is a very large polynomial of certain problem parameters). Furthermore, we consider a 3-layer convolutional network with max-pooling which is not studied in these works.

Soltanolkotabi et al. (2018), Du & Lee (2018) and Li et al. (2017) study the role of over-parameterization in the case of quadratic activation functions. Brutzkus et al. (2018) provide generalization guarantees for over-parameterized networks with Leaky ReLU activations on linearly separable data. Neyshabur et al. (2018) prove generalization bounds for neural networks. However, these bounds are empirically vacuous for over-parameterized networks and they do not prove that networks found by optimization algorithms give low generalization bounds.

---

[2]See Definition 5.1 for a formal definition of detection of a pattern.

## 3   PROBLEM FORMULATION

We begin with some notations and definitions. Let $d \geq 4$ be an integer. We consider a classification problem in the space $\{\pm 1\}^{2d}$. Namely, the space of vectors of $2d$ coordinates where each coordinate can be $+1$ or $-1$. Given a vector $\boldsymbol{x} \in \{\pm 1\}^{2d}$, we consider its partition into $d$ sets of two coordinates as follows $\boldsymbol{x} = (\boldsymbol{x}_1, ..., \boldsymbol{x}_d)$ where $\boldsymbol{x}_i \in \{\pm 1\}^2$. We refer to each such $\boldsymbol{x}_i$ as a *pattern* in $\boldsymbol{x}$.

**Neural Architecture:**   We consider learning with the following three-layer neural net model. The first layer is a convolutional layer with non-overlapping filters and multiple channels, the second layer is max pooling and the third layer is a fully connected layer with $2k$ hidden neurons and weights fixed to values $\pm 1$. Formally, for an input $\boldsymbol{x} = (\boldsymbol{x}_1, ..., \boldsymbol{x}_d) \in \mathbb{R}^{2d}$ where $\boldsymbol{x}_i \in \mathbb{R}^2$, the output of the network is given by:

$$N_W(\boldsymbol{x}) = \sum_{i=1}^{k} \left[ \max_j \left\{ \sigma \left( \boldsymbol{w}^{(i)} \cdot \boldsymbol{x}_j \right) \right\} - \max_j \left\{ \sigma \left( \boldsymbol{u}^{(i)} \cdot \boldsymbol{x}_j \right) \right\} \right] \tag{1}$$

where $W \in \mathbb{R}^{2k \times 2}$ is the weight matrix whose rows are the $\boldsymbol{w}^{(i)}$ vectors followed by the $\boldsymbol{u}^{(i)}$ vectors, and $\sigma(x) = \max\{0, x\}$ is the ReLU activation applied element-wise. See Figure 4 for an illustration of this architecture.

**Remark 3.1.** *Because there are only 4 different patterns, the network is limited in terms of the number of different rules it can implement. Specifically, it is easy to show that its VC dimension is at most 15 (see Sec. 10). Despite this limited expressive power, there is a generalization gap between small and large networks in this setting, as can be seen in Figure 1, and in our analysis below.*

**Data Generating Distribution:**   Next we define the classification rule we will focus on. Let $P_{XOR}$ correspond to the following two patterns: $P_{XOR} = \{(1,1), (-1,-1)\}$. Define the classification rule:

$$f^*(\boldsymbol{x}) = \begin{cases} 1 & \exists i \in \{1, \ldots, d\} : \boldsymbol{x}_i \in P_{XOR} \\ -1 & \text{otherwise} \end{cases} \tag{2}$$

Namely, $f^*$ detects whether a pattern in $P_{XOR}$ appears in the input. In what follows, we refer to $P_{XOR}$ as the set of positive patterns and $\{\pm 1\}^2 \setminus P_{XOR}$ as the set of negative patterns.

Let $\mathcal{D}$ be a distribution over $\mathcal{X} \times \{\pm 1\}$ such that for all $(\boldsymbol{x}, y) \sim \mathcal{D}$ we have $y = f^*(\boldsymbol{x})$. We say that a point $(\boldsymbol{x}, y)$ is positive if $y = 1$ and negative otherwise. Let $\mathcal{D}_+$ be the marginal distribution over $\{\pm 1\}^{2d}$ of positive points and $\mathcal{D}_-$ be the marginal distribution of negative points.

In the following definition we introduce the notion of *diverse* points, which will play a key role in our analysis.

**Definition 3.2 (Diverse Points).** *We say that a positive point $(\boldsymbol{x}, 1)$ is diverse if for all $\boldsymbol{z} \in \{\pm 1\}^2$ there exists $1 \leq i \leq d$ such that $\boldsymbol{x}_i = \boldsymbol{z}$. We say that a negative point $(\boldsymbol{x}, -1)$ is diverse if for all $\boldsymbol{z} \in \{\pm 1\}^2 \setminus P_{XOR}$ there exists $1 \leq i \leq d$ such that $\boldsymbol{x}_i = \boldsymbol{z}$.*

For $\phi \in \{-, +\}$ define $p_\phi$ to be the probability that $\boldsymbol{x}$ is diverse with respect to $\mathcal{D}_\phi$. For example, if both $D_+$ and $D_-$ are uniform, then by the inclusion-exclusion principle it follows that $p_+ = 1 - \frac{4 \cdot 3^d - 6 \cdot 2^d + 4}{4^d}$ and $p_- = 1 - \frac{1}{2^{d-1}}$.

For each set of binary patterns $A \subseteq \{\pm 1\}^2$ define $p_A$ to be the probability to sample a point which contains all patterns in $A$ and no patterns in $A^c$ (the complement of $A$). Let $A_1 = \{2\}$, $A_2 = \{4\}$, $A_3 = \{2, 4, 1\}$ and $A_4 = \{2, 4, 3\}$. The following quantity will be useful in our analysis:

$$p^* = \min_{1 \leq i \leq 4} p_{A_i} \tag{3}$$

**Learning Setup:**   Our analysis will focus on the problem of learning $f^*$ from training data with a three layer neural net model. The learning algorithm will be gradient descent, randomly initialized. As in any learning task in practice, $f^*$ is unknown to the training algorithm. Our goal is to analyze the performance of gradient descent when given data that is labeled with $f^*$. We assume that we are given a training set $S = S_+ \cup S_- \subseteq \{\pm 1\}^{2d} \times \{\pm 1\}^2$ where $S_+$ consists of $m$ IID points drawn from $\mathcal{D}_+$ and $S_-$ consists of $m$ IID points drawn from $\mathcal{D}_-$.[3]

---

[3]For simplicity, we consider this setting of equal number of positive and negative points in the training set.

Importantly, we note that the function $f^*$ can be realized by the above network with $k = 2$. Indeed, the network $N$ defined by the filters $\boldsymbol{w}^{(1)} = (3,3)$, $\boldsymbol{w}^{(2)} = (-3,-3)$, $\boldsymbol{u}^{(1)} = (-1,1)$, $\boldsymbol{u}^{(2)} = (1,-1)$ satisfies $\text{sign}(N(\boldsymbol{x})) = f^*(\boldsymbol{x})$ for all $\boldsymbol{x} \in \{\pm 1\}^{2d}$. It can be seen that for $k = 1$, $f^*$ cannot be realized. Therefore, any $k > 2$ is an over-parameterized setting.

**Training Algorithm:**    We will use gradient descent to optimize the following hinge-loss function.

$$\ell(W) = \frac{1}{m} \sum_{(\boldsymbol{x}_i, y_i) \in S_+ : y_i = 1} \max\{\gamma - N_W(\boldsymbol{x}_i), 0\} + \frac{1}{m} \sum_{(\boldsymbol{x}_i, y_i) \in S_- : y_i = -1} \max\{1 + N_W(\boldsymbol{x}_i), 0\} \quad (4)$$

for $\gamma \geq 1$. [4]We assume that gradient descent runs with a constant learning rate $\eta$ and the weights are randomly initiliazed with IID Gaussian weights with mean 0 and standard deviation $\sigma_g$. Furthermore, only the weights of the first layer, the convolutional filters, are trained. [5]

We will need the following notation. Let $W_t$ be the weight matrix in iteration $t$ of gradient descent. For $1 \leq i \leq k$, denote by $\boldsymbol{w}_t^{(i)} \in \mathbb{R}^2$ the $i$th convolutional filter at iteration $t$. Similarly, for $1 \leq i \leq k$ we define $\boldsymbol{u}_t^{(i)} \in \mathbb{R}^2$ to be the $k+i$ convolutional filter at iteration $t$. We assume that each $\boldsymbol{w}_0^{(i)}$ and $\boldsymbol{u}_0^{(i)}$ is initialized as a Gaussian random variable where the entries are IID and distributed as $\mathcal{N}(0, \sigma_g^2)$. In each iteration, gradient descent performs the update $W_{t+1} = W_t - \eta \frac{\partial \ell}{\partial W}(W_t)$.

## 4    MAIN RESULT

In this section we state our main result that demonstrates the generalization gap between over-parameterized networks and networks with $k = 2$. Define the generalization error to be the difference between the 0-1 test error and the 0-1 training error. For any $\epsilon$, $\delta$ and training algorithm let $m(\epsilon, \delta)$ be the sample complexity of a training algorithm, namely, the number of minimal samples the algorithm needs to get at most $\epsilon$ generalization error with probability at least $1 - \delta$. We consider running gradient descent in two cases, when $k \geq 120$ and $k = 2$. In the next section we exactly define under which set of parameters gradient descent runs, e.g., which constant learning rates.

Fix parameters $p_+$ and $p_-$ of a distribution $\mathcal{D}$ and denote by $c < 10^{-10}$ a negligible constant. Assume that gradient descent is given a sample of points drawn from $\mathcal{D}_+$ and $\mathcal{D}_-$. We denote the sample complexity of gradient descent in the cases $k \geq 120$ and $k = 2$, by $m_1$ and $m_2$, respectively. The following result shows a data dependent generalization gap (recall the definition of $p^*$ in Eq. 3).

**Theorem 4.1.** *Let $\mathcal{D}$ be a distribution with paramaters $p_+$, $p_-$ and $p^*$. Let $\delta \geq 1 - p_+ p_- (1 - c - 16e^{-8})$ and $0 \leq \epsilon < p^*$. Then $m_1(\epsilon, \delta) \leq 2$ whereas $m_2(\epsilon, \delta) \geq \frac{2 \log\left(\frac{48\delta}{33(1-c)}\right)}{\log(p_+ p_-)}$.* [6]

The proof follows from Theorem 5.2 and Theorem 5.3 which we state in the next section. The proof is given in Sec. 8.8. One surprising fact of this theorem is that $m_1(0, \delta) \leq 2$. Indeed, our analysis shows that for an over-parameterized network and for sufficiently large $p_+$ and $p_-$, one diverse positive point and one diverse negative suffice for gradient descent to learn $f^*$ with high probability. We note that even in this case, the dynamics of gradient descent is highly complex. This is due to the randomness of the initialization and to the fact that there are multiple weight filters in the network, each with different dynamics. See Sec. 5 for further details.

We will illustrate the guarantee of Theorem 4.1 with several numerical examples. In all of the examples we assume that for the distribution $\mathcal{D}$, the probability to sample a positive point is $\frac{1}{2}$ and

---

[4]In practice it is common to set $\gamma$ to 1. In our analyis we will need $\gamma \geq 8$ to guarantee generalization. In Section 8.3 we show empirically, that for this task, setting $\gamma$ to be larger than 1 results in better test performance than setting $\gamma = 1$.

[5]Note that Hoffer et al. (2018) show that fixing the last layer weights to $\pm 1$ does not degrade performance in various tasks. This assumption also appeared in other works (Brutzkus et al., 2018; Li & Yuan, 2017).

[6]We note that this generalization gap holds for global minima (0 train error). Therefore, the theorem can be read as follows. For $k \geq 120$, given 2 samples, with probability at least $1 - \delta$, gradient descent converges to a global minimum with at most $\epsilon$ test error. On the other hand, for $k = 2$ and given number of samples less than $\frac{2 \log\left(\frac{48\delta}{33(1-c)}\right)}{\log(p_+ p_-)}$, with probability greater than $\delta$, gradient descent converges to a global minimum with error greater than $\epsilon$. See Section 5 for further details.

$p^* = \min\left\{\frac{1-p_+}{4}, \frac{1-p_-}{4}\right\}$ (it is easy to constuct such distributions). In the first example, we assume that $p_+ = p_- = 0.98$ and $\delta = 1 - 0.98^2(1 - c - 16e^{-8}) \leq 0.05$. In this case we get that for any $0 \leq \epsilon < 0.005$, $m_1(\epsilon, \delta) \leq 2$ whereas $m_2(\epsilon, \delta) \geq 129$. For the second example consider the case where $p_+ = p_- = 0.92$. It follows that for $\delta = 0.16$ and any $0 \leq \epsilon < 0.02$ it holds that $m_1(\epsilon, \delta) \leq 2$ and $m_2(\epsilon, \delta) \geq 17$.

For $\epsilon = 0$ and any $\delta > 0$, by setting $p_+$ and $p_-$ to be sufficiently close to 1, we can get an arbitrarily large gap between $m_1(\epsilon, \delta)$ and $m_2(\epsilon, \delta)$. In contrast, for sufficiently small $p_+$ and $p_-$, e.g., in which $p_+, p_- \leq 0.7$, our bound does not guarantee a generalization gap.

## 5 PROOF SKETCH AND INSIGHTS

In this section we sketch the proof of Theorem 4.1. The theorem follows from two theorems: Theorem 5.2 for over-parameterized networks and Theorem 5.3 for networks with $k = 2$. We formally show this in Sec. 8.8. In Sec. 5.1 we state Theorem 5.2 and outline its proof. In Sec. 5.2 we state Theorem 5.3 and shortly outline its proof. Finally, for completeness, in Sec. 9 we also provide a convergence guarantee for the XOR problem with inputs in $\{\pm 1\}$, which in our setting is the case of $d = 1$. In what follows, we will need the following formal definition for a detection of a pattern by a network.

**Definition 5.1.** *Let* $c_d \geq 0$ *be a constant. For each positive pattern* $\boldsymbol{v}_p$ *define* $D_{\boldsymbol{v}_p} = \sum_{i=1}^{k} \sigma\left(\boldsymbol{w}^{(i)} \cdot \boldsymbol{v}_p\right)$ *and for each negative pattern* $\boldsymbol{v}_n$ *define* $D_{\boldsymbol{v}_n} = \sum_{i=1}^{k} \sigma\left(\boldsymbol{u}^{(i)} \cdot \boldsymbol{v}_n\right)$. *We say that a pattern* $\boldsymbol{v}$ *(positive or negative) is detected by the network* $N_W$ *with confidence* $c_d$ *if* $D_{\boldsymbol{v}} > c_d$.

The above definition captures a desired property of a network, namely, that its filters which are connected with a positive coefficient in the last layer, have high correlation with the positive patterns and analogously for the remaining filters and negative patterns. We note however, that the condition in which a network detects all patterns is not equivalent to realizing the ground truth $f^*$. The former can hold without the latter and vice versa.

Theorem 5.2 and Theorem 5.3 together imply a clear characterization of the different inductive biases of gradient descent in the case of small ($k = 2$) and over-parameterized networks. The characterization is that over-parameterized networks are biased towards global minima that detect all patterns in the data, whereas small networks with $k = 2$ are biased towards global minima that do not detect all patterns (see Definition 5.1). In Sec. 8.5 we show this empirically in the XORD problem and in a generalization of the XORD problem.

In the following sections we will need several notations. Define $\boldsymbol{x}_1 = (1, 1), \boldsymbol{x}_2 = (1, -1), \boldsymbol{x}_3 = (-1, -1), \boldsymbol{x}_4 = (-1, 1)$ to be the four possible patterns in the data and the following sets:

$$W_t^+(i) = \left\{j \mid \arg\max_{1 \leq l \leq 4} \boldsymbol{w}_t^{(j)} \cdot \boldsymbol{x}_l = i\right\}, \quad U_t^+(i) = \left\{j \mid \arg\max_{1 \leq l \leq 4} \boldsymbol{u}_t^{(j)} \cdot \boldsymbol{x}_l = i\right\}$$

$$W_t^-(i) = \left\{j \mid \arg\max_{l \in \{2,4\}} \boldsymbol{w}_t^{(j)} \cdot \boldsymbol{x}_l = i\right\}, \quad U_t^-(i) = \left\{j \mid \arg\max_{l \in \{2,4\}} \boldsymbol{u}_t^{(j)} \cdot \boldsymbol{x}_l = i\right\} \qquad (5)$$

We denote by $\boldsymbol{x}^+$ a positive diverse point and $\boldsymbol{x}^-$ a negative diverse point. Define the following sum:

$$S_t^+ = \sum_{j \in W_t^+(1) \cup W_t^+(3)} \left[\max\left\{\sigma\left(\boldsymbol{w}^{(j)} \cdot \boldsymbol{x}_1^+\right), ..., \sigma\left(\boldsymbol{w}^{(j)} \cdot \boldsymbol{x}_d^+\right)\right\}\right]$$

Finally, in all of the results in this section we will denote by $c < 10^{-10}$ a negligible constant.

### 5.1 SAMPLE COMPLEXITY UPPER BOUND FOR OVER-PARAMETERIZED NETWORKS

The main result in this section is given by the following theorem.

**Theorem 5.2.** *Let* $S = S_+ \cup S_-$ *be a training set as in Sec. 3. Assume that gradient descent runs with parameters* $\eta = \frac{c_\eta}{k}$ *where* $c_\eta \leq \frac{1}{410}$, $\sigma_g \leq \frac{c_\eta}{16k^{\frac{3}{2}}}$, $k \geq 120$ *and* $\gamma \geq 8$. *Then, with probability*

*at least* $(p_+ p_-)^m \left(1 - c - 16e^{-8}\right)$ *after running gradient descent for* $T \geq \frac{28(\gamma + 1 + 8c_\eta)}{c_\eta}$ *iterations, it converges to a global minimum which satisfies:*

1. $\mathrm{sign}\left(N_{W_T}(\boldsymbol{x})\right) = f^*(\boldsymbol{x})$ *for all* $\boldsymbol{x} \in \{\pm 1\}^{2d}$.

2. *Let* $\alpha(k) = \frac{\frac{k}{4} + 2\sqrt{k}}{\frac{k}{4} - 2\sqrt{k}}$. *Then for* $c_d \leq \frac{1 - \frac{5c_\eta}{4}}{\alpha(k)+1}$, *all patterns are detected with confidence* $c_d$.

This result shows that given a small training set size, and sufficiently large $p_+$ and $p_-$, over-parameterized networks converge to a global minimum which realizes the classifier $f^*$ with high probability and in a constant number of iterations. Furthermore, this global minimum detects all patterns in the data with confidence that increases with over-parameterization. The full proof of Theorem 5.2 is given in Sec. 8.6.

We will now sketch its proof. With probability at least $(p_+ p_-)^m$ all training points are diverse and we will condition on this event. From Sec. 10 we can assume WLOG that the training set consists of one positive diverse point $\boldsymbol{x}^+$ and one negative diverse point $\boldsymbol{x}^-$ (since the network will have the same output on all same-label diverse points). We note that empirically over-parameterization improves generalization even when the training set contains non-diverse points (see Fig. 1 and Sec. 8.2).

Now, to understand the dynamics of gradient descent it is crucial to understand the dynamics of the sets in Eq. 5. This follows since the gradient updates are expressed via these sets. Concretely, let $j \in W_t^+(i_1) \cap W_t^-(i_2)$ then the gradient update is given as follows: [7]

$$\boldsymbol{w}_{t+1}^{(j)} = \boldsymbol{w}_t^{(j)} + \eta \boldsymbol{x}_{i_1} \mathbb{1}_{N_W(\boldsymbol{x}^+) < \gamma} - \eta \boldsymbol{x}_{i_2} \mathbb{1}_{N_W(\boldsymbol{x}^-) < 1} \tag{6}$$

Similarly, for $j \in U_t^+(i_1) \cap U_t^-(i_2)$ the gradient update is given by:

$$\boldsymbol{u}_{t+1}^{(j)} = \boldsymbol{u}_t^{(j)} - \eta \boldsymbol{x}_{i_1} \mathbb{1}_{N_W(\boldsymbol{x}^+) < \gamma} + \eta \boldsymbol{x}_{i_2} \mathbb{1}_{N_W(\boldsymbol{x}^-) < 1} \tag{7}$$

Furthermore, the values of $N_W(\boldsymbol{x}^+)$ and $N_W(\boldsymbol{x}^-)$ depend on these sets and their corresponding weight vectors, via sums of the form $S_t^+$, defined above.

The proof consists of a careful analysis of the dynamics of the sets in Eq. 5 and their corresponding weight vectors. For example, one result of this analysis is that for all $t \geq 1$ and $i \in \{1, 3\}$ we have $W_t^+(i) = W_0^+(i)$ and the size of $W_t^+(i)$ is at least $\frac{k}{2} - 2\sqrt{k}$ with high probability.

There are two key technical observations that we apply in this analysis. First, with a small initialization and with high probability, for all $1 \leq j \leq k$ and $1 \leq i \leq 4$ it holds that $\left| \boldsymbol{w}_0^{(j)} \cdot \boldsymbol{x}_i \right| \leq \frac{\eta}{4}$ and $\left| \boldsymbol{u}_0^{(j)} \cdot \boldsymbol{x}_i \right| \leq \frac{\eta}{4}$. This allows us to keep track of the dynamics of the sets in Eq. 5 more easily. For example, by this observation it follows that if for some $j^* \in W_t^+(2)$ it holds that $j^* \in W_{t+1}^+(4)$, then *for all* $j$ such that $j \in W_t^+(2)$ it holds that $j \in W_{t+1}^+(4)$. Hence, we can reason about the dynamics of several filters all at once, instead of each one separately. Second, by concentration of measure we can estimate the sizes of the sets in Eq. 5 at iteration $t = 0$. Combining this with results of the kind $W_t^+(i) = W_0^+(i)$ for all $t$, we can understand the dynamics of these sets throughout the optimization process.

The theorem consists of optimization and generalization guarantees. For the optimization guarantee we show that gradient descent converges to a global minimum. To show this, the idea is to characterize the dynamics of $S_t^+$ using the characterization of the sets in Eq. 5 and their corresponding weight vectors. We show that as long as gradient descent did not converge to a global minimum, $S_t^+$ cannot decrease in any iteration and it is upper bounded by a constant. Furthermore, we show that there cannot be too many consecutive iterations in which $S_t^+$ does not increase. Therefore, after sufficiently many iterations gradient descent will converge to a global minimum.

We will now outline the proof of the generalization guarantee. Denote the network learned by gradient descent by $N_{W_T}$. First, we show that the network classifies all positive points correctly.

---

[7]Note that with probability 1, $\sigma'(\boldsymbol{w}_t^{(j)} \cdot \boldsymbol{x}_{i_1}) = 1$, $\sigma'(\boldsymbol{w}_t^{(j)} \cdot \boldsymbol{x}_{i_2}) = 1$ for all $t$, and therefore we omit these from the gradient update. This follows since $\sigma'(\boldsymbol{w}_t^{(j)} \cdot \boldsymbol{x}_{i_1}) = 0$ for some $t$ only if $\boldsymbol{w}_0^{(j)} \cdot \boldsymbol{x}_{i_1}$ is an integer multiple of $\eta$.

Define the following sums for all $1 \leq i \leq 4$:

$$X_t^+(i) = \sum_{j \in W_T^+(i)} \max_k \sigma\left(\boldsymbol{w}^{(j)} \cdot \boldsymbol{x}_k^+\right) \ , \ \ Y_t^+(i) = \sum_{j \in U_T^+(i)} \max_k \sigma\left(\boldsymbol{u}^{(j)} \cdot \boldsymbol{x}_k^+\right) \tag{8}$$

First we notice that for all positive $\boldsymbol{z}$ we have $N_{W_T}(\boldsymbol{z}) \gtrsim \min\{X_T^+(1), X_T^+(3)\} - Y_T^+(2) - Y_T^+(4)$. Then by the fact that $N_{W_T}(\boldsymbol{x}^+) \geq \gamma$ at the global minimum, we can show that $X_T^+(1) + X_T^+(3)$ is sufficiently large. As mentioned previously, by concentration of measure, $\left|W_0^+(i)\right|$ is sufficiently large. Then by a symmetry argument we show that this implies that both $X_T^+(1)$ and $X_T^+(3)$ are sufficiently large. This shows that patterns $\boldsymbol{x}_1$ and $\boldsymbol{x}_3$ are detected. Finally, we show that $Y_T^+(2) + Y_T^+(4)$ is not too large due to an upper bound on $-N_{W_T}(\boldsymbol{x}^-)$. Hence, we can show that each positive point is classified correctly. The proof that all negative points are classified correctly and patterns $\boldsymbol{x}_2$ and $\boldsymbol{x}_4$ are detected is similar but slightly more technical. We refer the reader to Sec. 8.6 for further details.

## 5.2 SAMPLE COMPLEXITY LOWER BOUND FOR SMALL NETWORKS ($k = 2$)

The following theorem provides generalization lower bounds of global minima in the case that $k = 2$ and in a slightly more general setting than the one given in Theorem 5.2.

**Theorem 5.3.** *Let $S = S_+ \cup S_-$ be a training set as in Sec. 3. Assume that gradient descent runs with parameters $\eta = \frac{c_\eta}{k}$ where $c_\eta \leq \frac{1}{41}$, $\sigma_g \leq \frac{c_\eta}{16k^{\frac{3}{2}}}$, $k = 2$ and $\gamma \geq 1$. Then the following holds:*

1. *With probability at least $(p_+ p_-)^m (1 - c) \frac{33}{48}$, gradient descent converges to a global minimum that has non-zero test error. Furthermore, for $c_d \geq 2c_\eta$, there exists at least one pattern which is not detected by the global minimum with confidence $c_d$.*

2. *The non-zero test error above is at least $p^*$.*

The theorem shows that for a training set that is not too large and given sufficiently large $p_+$ and $p_-$, with constant probability, gradient descent will converge to a global minimum that is not the classifier $f^*$. Furthermore, this global minimum does not detect at least one pattern. The proof of the theorem is given in Sec. 8.7.

We will now provide a short outline of the proof. Let $\boldsymbol{w}_T^{(1)}$, $\boldsymbol{w}_T^{(2)}$, $\boldsymbol{u}_T^{(1)}$ and $\boldsymbol{u}_T^{(2)}$ be the filters of the network at the iteration $T$ in which gradient descent converges to a global minimum. The proof shows that gradient descent will not learn $f^*$ if one of the following conditions is met: a) $W_T^+(1) = \emptyset$. b) $W_T^+(3) = \emptyset$. c) $\boldsymbol{u}_T^{(1)} \cdot \boldsymbol{x}_2 > 0$ and $\boldsymbol{u}_T^{(2)} \cdot \boldsymbol{x}_2 > 0$. d) $\boldsymbol{u}_T^{(1)} \cdot \boldsymbol{x}_4 > 0$ and $\boldsymbol{u}_T^{(2)} \cdot \boldsymbol{x}_4 > 0$. Then by using a symmetry argument which is based on the symmetry of the initialization and the training data it can be shown that one of the above conditions is met with high constant probability. Finally, it can be shown that if one of these conditions hold, then at least one pattern is not detected.

## 6 EXPERIMENTS

We perform several experiments that corroborate our theoretical findings. In Sec. 8.5 we empirically demonstrate our insights on the inductive bias of gradient descent. In Sec. 6.2 we evaluate a model compression scheme implied by our results, and demonstrate its success on the MNIST dataset.

## 6.1 PATTERN DETECTION

In this section we perform experiments to examine the insights from our analysis on the inductive bias of gradient descent. Namely, that over-parameterized networks are biased towards global minima that detect more patterns in the data than global minima found by smaller networks. We check this both on the XORD problem which contains 4 possible patterns in the data and on an instance of an extension of the XORD problem, that we refer to as the Orthonormal Basis Detection (OBD) problem, which contains 60 patterns in the data. In Sec. 8.5 we provide details on the experimental setups.

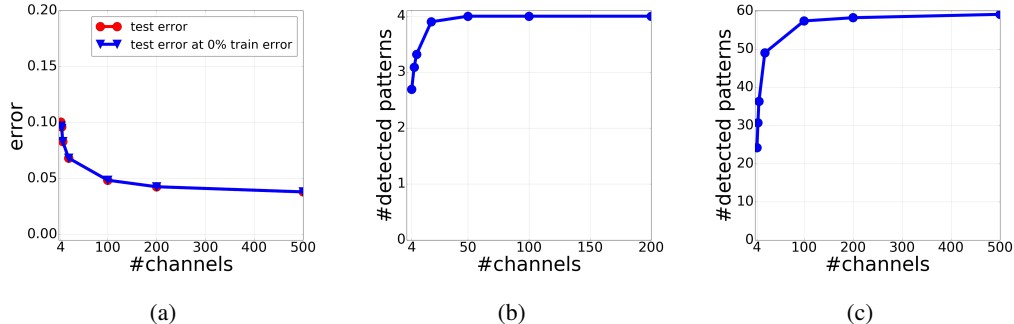

Figure 2: Inductive bias in XORD and OBD problems. (a) Over-parameterization improves generalization in the OBD problem (b) Pattern detection phenomenon in the XORD problem (c) Pattern detection phenomenon in the OBD problem. In both (b) and (c) we see that as the number of channels increase, gradient descent is biased towards %0 training error solutions with more detected patterns.

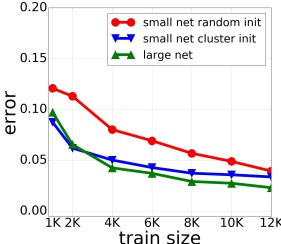

Figure 3: Model compression in MNIST. The plot shows the test error of the small network (4 channels) with standard training (red), the small network that uses clusters from the large network (blue), and the large network (120 channels) with standard training (green). It can be seen that the large network is effectively compressed without losing much accuracy.

Due to space considerations, we will not formally define the OBD problem in this section. We refer the reader to Sec. 8.5 for a formal definition. Informally, The OBD problem is a natural extension of the XORD problem that contains more possible patterns in the data and allows the dimension of the filters of the convolutional network to be larger. The patterns correspond to a set of orthonormal vectors and their negations. The ground truth classifier in this problem can be realized by a convolutional network with 4 channels.

In Fig. 2 we show experiments which confirm that in the OBD problem as well, overparameterization improves generalization. We further show the number of patterns detected in %0 training error solutions for different number of channels, in both the XORD and OBD problems. It can be clearly seen that for both problems, over-parameterized networks are biased towards %0 training error solutions that detect more patterns, as predicted by the theoretical results.

## 6.2 NETWORK COMPRESSION

By inspecting the proof of Theorem 5.2, one can see that the dynamics of the filters of an over-parameterized network are such that they either have low norm, or they have large norm and they point to the direction of one of the patterns (see, e.g., Lemma 8.4 and Lemma 8.6). This suggests that by clustering the filters of a trained over-parameterized network to a small number of clusters, one can create a significantly smaller network which contains all of the detectors that are needed for good generalization performance. Then, by training the last layer of the network, it can converge to a good solution. Following this insight, we tested this procedure on the MNIST data set and a 3 layer convolutional network with convolutional layer with multiple channels and $3 \times 3$ kernels, max pooling layer and fully connected layer. We trained an over-parameterized network with 120 channels, clustered its filters with k-means into 4 clusters and used the cluster centers as initialization for a small network with 4 channels. Then we trained only the fully connected layer of the small

network. In Fig. 3 we show that for various training set sizes, the performance of the small network improves significantly with the new initialization and nearly matches the performance of the over-parameterized network.

## 7    CONCLUSIONS

In this paper we consider a simplified learning task on binary vectors and show that over-parameterization can provably improve generalization performance of a 3-layer convolutional network trained with gradient descent. Our analysis reveals that in the XORD problem over-parameterized networks are biased towards global minima which detect more relevant patterns in the data. While we prove this only for the XORD problem and under the assumption that the training set contains diverse points, our experiments clearly show that a similar phenomenon occurs in other settings as well. We show that this is the case for XORD with non-diverse points (Figure 1) and in the more general OBD problem which contains 60 patterns in the data and is not restricted to binary inputs (Figure 2). Furthermore, our experiments on MNIST hint that this is the case in MNIST as well (Figure 3).By clustering the detected patterns of the large network we could achieve better accuracy with a small network. This suggests that the larger network detects more patterns with gradient descent even though its effective size is close to that of a small network.

We believe that these insights and our detailed analysis can guide future work for showing similar results in more complex tasks and provide better understanding of this phenomenon. It would also be interesting to further study the implications of such results on model compression and on improving training algorithms.

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

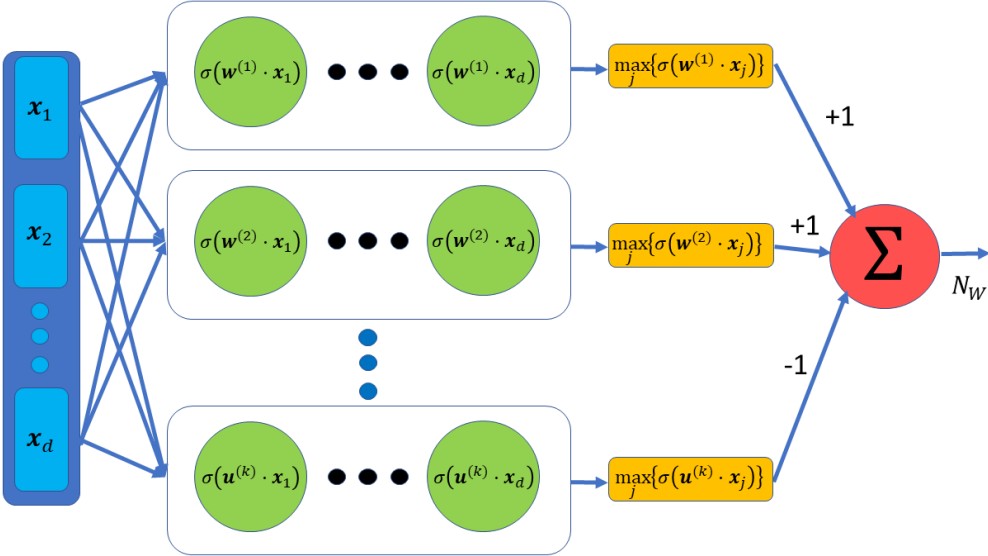

Figure 4: Network architecture used for training in the XORD problem.

Behnam Neyshabur, Zhiyuan Li, Srinadh Bhojanapalli, Yann LeCun, and Nathan Srebro. Towards understanding the role of over-parametrization in generalization of neural networks. *arXiv preprint arXiv:1805.12076*, 2018.

Roman Novak, Yasaman Bahri, Daniel A Abolafia, Jeffrey Pennington, and Jascha Sohl-Dickstein. Sensitivity and generalization in neural networks: an empirical study. *arXiv preprint arXiv:1802.08760*, 2018.

Mahdi Soltanolkotabi, Adel Javanmard, and Jason D Lee. Theoretical insights into the optimization landscape of over-parameterized shallow neural networks. *IEEE Transactions on Information Theory*, 2018.

Ida G Sprinkhuizen-Kuyper and Egbert JW Boers. The error surface of the 2-2-1 xor network: The finite stationary points. *Neural Networks*, 11(4):683–690, 1998.

Roman Vershynin. High-dimensional probability. *An Introduction with Applications*, 2017.

# 8 APPENDIX

## 8.1 NETWORK ARCHITECTURE IN THE XORD PROBLEM

## 8.2 EXPERIMENTAL SETUPS

### 8.2.1 EXPERIMENT IN FIGURE 1

We tested the generalization performance in the setup of Section3. We considered networks with number of channels 4,6,8,20,50,100 and 200. The distribution in this setting has $p_+ = 0.5$ and $p_- = 0.9$ and the training sets are of size 12 (6 positive, 6 negative). Note that in this case the training set contains non-diverse points with high probability. The ground truth network can be realized by a network with 4 channels. For each number of channels we trained a convolutional network 100 times and averaged the results. In each run we sampled a new training set and new initialization of the weights according to a gaussian distribution with mean 0 and standard deviation 0.00001. For each number of channels $c$, we ran gradient descent with learning rate $\frac{0.04}{c}$ and stopped it if it did not improve the cost for 20 consecutive iterations or if it reached 30000 iterations. The last iteration was taken for the calculations. We plot both average test error over all 100 runs and

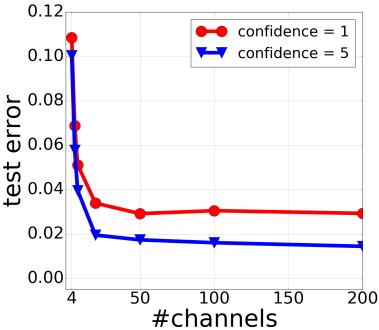

Figure 5: Higher confidence of hinge-loss results in better performance in the XORD problem.

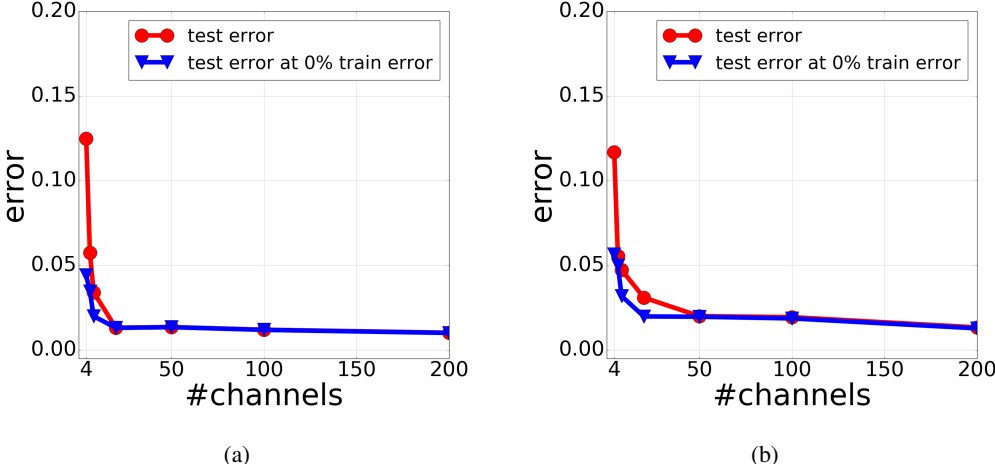

(a)                                           (b)

Figure 6: Demonstration of generalization gap for values not included in Theorem 5.3. The experimental setup is the same as in Section 8.2.1 with the following exceptions. For all number of channels we changed the standard devation $\sigma_g$ and only for $k = 2$ we changed the learning rate $\eta$, as described next for each subfigure. Furthermore, the number of runs is 40 for each channel (instead of 100). (a) Experiments for $\eta = 0.1$ and $\sigma_g = 0.01$. (b) Experiments for $\eta = 0.1$ and $\sigma_g = 0.1$. Finally, we note that there is a genereliazation gap for gradient descent when comparing the performance for $k = 2$ in these experiments and for larger $k$ in the experiments in Figure 1.

average test error only over the runs that ended at 0% train error. In this case, for each number of channels 4,6,8,20,50,100,200 the number of runs in which gradient descent converged to a 0% train error solution is 62, 79, 94, 100, 100, 100, 100, respectively.

## 8.3   HINGE LOSS CONFIDENCE

Figure 5 shows that setting $\gamma = 5$ gives better performance than setting $\gamma = 1$ in the XORD problem. The setting is similar to the setting of Section 8.2.1. Each point is an average test error of 100 runs.

## 8.4   EXPERIMENTS FOR SECTION 5.2

Theorem 5.3 holds for any $\eta \leq \frac{1}{82}$ and $\sigma \leq \frac{1}{1900}$. Because the result is a lower bound, it is desirable to understand the behaviour of gradient descent for values outside these ranges. In Figure 6 we empirically show that for values outside these ranges, there is a generalization gap between gradient descent for $k = 2$ and gradient descent for larger $k$.

## 8.5 EXPERIMENTAL SETUP IN SECTION 6

We will first formally define the OBD problem. Fix an even dimension parameter $d_1 \geq 2$. In this problem, we assume there is an orthonormal basis $B = \{\boldsymbol{v}_1, ..., \boldsymbol{v}_{d_1}\}$ of $\mathbb{R}^{d_1}$. Divide $B$ into two equally sized sets $B_1$ and $B_2$, each of size $\frac{d_1}{2}$. Now define the set of positive patterns to be $P = \{\boldsymbol{v} \mid \boldsymbol{v} \in B_1\} \cup \{-\boldsymbol{v} \mid \boldsymbol{v} \in B_1\}$ and negative patterns to be $N = \{\boldsymbol{v} \mid \boldsymbol{v} \in B_2\} \cup \{-\boldsymbol{v} \mid \boldsymbol{v} \in B_2\}$. Let $P_{OBD} = P \cup N$. For $d_2 > 0$, we assume the input domain is $\mathcal{X} \subseteq \mathbb{R}^{d_1 d_2}$ and each $\boldsymbol{x} \in \mathcal{X}$ is a vector such that $\boldsymbol{x} = (\boldsymbol{x}_1, ..., \boldsymbol{x}_{d_2})$ where each $\boldsymbol{x}_i \in P_{OBD}$. We define the ground truth classifier $f_{OBD} : \mathcal{X} \to \{\pm 1\}$ such that $f_{OBD}(\boldsymbol{x}) = 1$ if and only there exists at least one $\boldsymbol{x}_i$ such that $\boldsymbol{x}_i \in P$. Notice that for $d_1 = 2$ and by normalizing the four vectors in $\{\pm 1\}^2$ to have unit norm, we get the XORD problem. We note that the positive patterns in the XORD problem are defined to be $P_{XOR}$ and the negative patterns are $\{\pm 1\}^2 \setminus P_{XOR}$.

Let $\mathcal{D}$ be a distribution over $\mathcal{X}^{2d} \times \{\pm 1\}$ such that for all $(\boldsymbol{x}, y) \sim \mathcal{D}$, $y = f_{OBD}(\boldsymbol{x})$. As in the XORD problem we define the distributions $\mathcal{D}_+$ and $\mathcal{D}_-$. We consider the following learning task which is the same as the task for the XORD problem. We assume that we are given a training set $S = S_+ \cup S_- \subseteq \{\pm 1\}^{d_1 d_2} \times \{\pm 1\}$ where $S_+$ consists of $m$ IID points drawn from $\mathcal{D}_+$ and $S_-$ consists of $m$ IID points drawn from $\mathcal{D}_-$. The goal is to train a neural network with randomly initialized gradient descent on $S$ and obtain a network $N : \mathbb{R}^{d_1 d_2} \to \mathbb{R}$ such that $\text{sign}(N(\boldsymbol{x})) = f_{OBD}(\boldsymbol{x})$ for all $\boldsymbol{x} \in \{\pm 1\}^{d_1 d_2}$.

We consider the same network as in the XORD problem (Eq. 1), but now the filters of the convolution layer are $d_1$-dimensional. Formally, for an input $\boldsymbol{x} = (\boldsymbol{x}_1, ..., \boldsymbol{x}_d) \in \mathcal{X}$ the output of the network is given by

$$
\begin{aligned}
N_W(\boldsymbol{x}) = \sum_{i=1}^{k} \Big[ & \max \left\{ \sigma\left(\boldsymbol{w}^{(i)} \cdot \boldsymbol{x}_1\right), ..., \sigma\left(\boldsymbol{w}^{(i)} \cdot \boldsymbol{x}_d\right) \right\} \\
& - \max \left\{ \sigma\left(\boldsymbol{u}^{(i)} \cdot \boldsymbol{x}_1\right), ..., \sigma\left(\boldsymbol{u}^{(i)} \cdot \boldsymbol{x}_d\right) \right\} \Big]
\end{aligned}
\tag{9}
$$

where $W \in \mathbb{R}^{2k \times d_1}$ is the weight matrix which contains in the first $k$ rows the vectors $\boldsymbol{w}^{(i)} \in \mathbb{R}^{d_1}$, in the next $k$ rows the vectors $\boldsymbol{u}^{(i)} \in \mathbb{R}^{d_1}$ and $\sigma(x) = \max\{0, x\}$ is the ReLU activation applied element-wise. We performed experiments in the case that $d_1 = 30$, i.e., in which there are 60 possible patterns.

In Figure 2a, for each number of channels we trained a convolutional network given in Eq. 9 with gradient descent for 100 runs and averaged the results. The we sampled 25 positive points and 25 negative points in the following manner. For each positive point we sampled with probability 0.25 one of the numbers [4,6,8,10] twice with replacement. Denote these numbers by $m_1$ and $m_2$. Then we sampled $m_1$ different positive patterns and $m_2$ different negative patterns. Then we filled a $60d_1$-dimensional vectors with all of these patterns. A similar procedure was used to sample a negative point. We considered networks with number of channels 4,6,8,20,100 and 200 and 500. Note that the ground truth network can be realized by a network with 4 channels. For each number of channels we trained a convolutional network 100 times and averaged the results. For each number of channels $c$, we ran gradient descent with learning rate $\frac{0.2}{c}$ and stopped it if it did not improve the cost for 20 consecutive iterations or if it had 0% training error for 200 consecutive iterations or if it reached 30000 iterations. The last iteration was taken for the calculations.

We plot both average test error over all 100 runs and average test error only over the runs that ended at 0% train error. For each number of channels 4,6,8,20,100,200,500 the number of runs in which gradient descent converged to a 0% train error solution is 96, 99, 100, 100, 100, 100, 100, respectively. For each 0% train error solution we recorded the number of patterns detected with $c_d = 0.0001$ according to the Definition 5.1 (generalized to the OBD problem). In the XORD problem we recorded similarly the number of patterns detected in experiments which are identical to the experiments in Section 8.2.1, except that in this case $p_+ = p_- = 0.98$.

## 8.6 PROOF OF THEOREM 5.2

We will first need a few notations. Define $\boldsymbol{x}_1 = (1,1), \boldsymbol{x}_2 = (1,-1), \boldsymbol{x}_3 = (-1,-1), \boldsymbol{x}_4 = (-1,1)$ and the following sets:

$$W_t^+(i) = \left\{ j \mid \arg\max_{1 \le l \le 4} \boldsymbol{w}_t^{(j)} \cdot \boldsymbol{x}_l = i \right\}, \ U_t^+(i) = \left\{ j \mid \arg\max_{1 \le l \le 4} \boldsymbol{u}_t^{(j)} \cdot \boldsymbol{x}_l = i \right\}$$

$$W_t^-(i) = \left\{ j \mid \arg\max_{l \in \{2,4\}} \boldsymbol{w}_t^{(j)} \cdot \boldsymbol{x}_l = i \right\}, \ U_t^-(i) = \left\{ j \mid \arg\max_{l \in \{2,4\}} \boldsymbol{u}_t^{(j)} \cdot \boldsymbol{x}_l = i \right\}$$

We can use these definitions to express more easily the gradient updates. Concretely, let $j \in W_t^+(i_1) \cap W_t^-(i_2)$ then the gradient update is given as follows:[8]

$$\boldsymbol{w}_{t+1}^{(j)} = \boldsymbol{w}_t^{(j)} + \eta \boldsymbol{x}_{i_1} \mathbb{1}_{N_W(\boldsymbol{x}^+) < \gamma} - \eta \boldsymbol{x}_{i_2} \mathbb{1}_{N_W(\boldsymbol{x}^-) < 1} \tag{10}$$

Similarly, for $j \in U_t^+(i_1) \cap U_t^-(i_2)$ the gradient update is given by:

$$\boldsymbol{u}_{t+1}^{(j)} = \boldsymbol{u}_t^{(j)} - \eta \boldsymbol{x}_{i_1} \mathbb{1}_{N_W(\boldsymbol{x}^+) < \gamma} + \eta \boldsymbol{x}_{i_2} \mathbb{1}_{N_W(\boldsymbol{x}^-) < 1} \tag{11}$$

We denote by $\boldsymbol{x}^+$ a positive diverse point and $\boldsymbol{x}^-$ a negative diverse point. Define the following sums for $\phi \in \{+,-\}$:

$$S_t^\phi = \sum_{j \in W_t^+(1) \cup W_t^+(3)} \left[ \max\left\{ \sigma\left(\boldsymbol{w}^{(j)} \cdot \boldsymbol{x}_1^\phi\right), ..., \sigma\left(\boldsymbol{w}^{(j)} \cdot \boldsymbol{x}_d^\phi\right) \right\} \right]$$

$$P_t^\phi = \sum_{j \in U_t^+(1) \cup U_t^+(3)} \left[ \max\left\{ \sigma\left(\boldsymbol{u}^{(j)} \cdot \boldsymbol{x}_1^\phi\right), ..., \sigma\left(\boldsymbol{u}^{(j)} \cdot \boldsymbol{x}_d^\phi\right) \right\} \right]$$

$$R_t^\phi = \sum_{j \in W_t^+(2) \cup W_t^+(4)} \left[ \max\left\{ \sigma\left(\boldsymbol{w}^{(j)} \cdot \boldsymbol{x}_1^\phi\right), ..., \sigma\left(\boldsymbol{w}^{(j)} \cdot \boldsymbol{x}_d^\phi\right) \right\} \right]$$
$$- \sum_{j \in U_t^+(2) \cup U_t^+(4)} \left[ \max\left\{ \sigma\left(\boldsymbol{u}^{(i)} \cdot \boldsymbol{x}_1^\phi\right), ..., \sigma\left(\boldsymbol{u}^{(i)} \cdot \boldsymbol{x}_d^\phi\right) \right\} \right]$$

Note that $R_t^+ = R_t^-$ since for $\boldsymbol{z} \in \{\boldsymbol{x}^+, \boldsymbol{x}^-\}$ there exists $i_1, i_2$ such that $\boldsymbol{z}_{i_1} = \boldsymbol{x}_2, \boldsymbol{z}_{i_2} = \boldsymbol{x}_4$.

By the conditions of the theorem, with probability at least $(p_+ p_-)^m$ all the points in the training set are diverse. From now on we will condition on this event. Furthermore, without loss of generality, we can assume that the training set consists of one diverse point $\boldsymbol{x}^+$ and one negative points $\boldsymbol{x}^-$. This follows since the network and its gradient have the same value for two different positive diverse points and two different negative points. Therefore, this holds for the loss function defined in Eq. 4 as well.

We will now proceed to prove the theorem. In Section 8.6.1 we prove results on the filters at initialization. In Section 8.6.2 we prove several auxiliary lemmas. In Section 8.6.3 we prove upper bounds on $S_t^-$, $P_t^+$ and $P_t^-$ for all iterations $t$. In Section 8.6.4 we characterize the dynamics of $S_t^+$ and in Section 8.6.5 we prove an upper bound on it together with upper bounds on $N_{W_t}(\boldsymbol{x}^+)$ and $-N_{W_t}(\boldsymbol{x}^-)$ for all iterations $t$.

We provide an optimization guarantee for gradient descent in Section 8.6.6. We prove generalization guarantees for the points in the positive class and negative class in Section 8.6.7 and Section 8.6.8, respectively. We complete the proof of the theorem in Section 8.6.9.

---

[8]Note that with probability 1, $\sigma'(\boldsymbol{w}_t^{(j)} \cdot \boldsymbol{x}_{i_1}) = 1$, $\sigma'(\boldsymbol{w}_t^{(j)} \cdot \boldsymbol{x}_{i_2}) = 1$ for all $t$, and therefore we omit these from the gradient update. This follows since $\sigma'(\boldsymbol{w}_t^{(j)} \cdot \boldsymbol{x}_{i_1}) = 0$ for some $t$ if and only if $\boldsymbol{w}_0^{(j)} \cdot \boldsymbol{x}_{i_1}$ is an integer multiple of $\eta$.

### 8.6.1 INITIALIZATION GUARANTEES

**Lemma 8.1.** *With probability at least $1 - 4e^{-8}$, it holds that*

$$\left| \left| W_0^+(1) \cup W_0^+(3) \right| - \frac{k}{2} \right| \leq 2\sqrt{k}$$

*and*

$$\left| \left| U_0^+(1) \cup U_0^+(3) \right| - \frac{k}{2} \right| \leq 2\sqrt{k}$$

*Proof.* Without loss of generality consider $\left| W_0^+(1) \cup W_0^+(3) \right|$. Since $\mathbb{P}\left[ j \in W_0^+(1) \cup W_0^+(3) \right] = \frac{1}{2}$, we get by Hoeffding's inequality

$$\mathbb{P}\left[ \left| \left| W_0^+(1) \cup W_0^+(3) \right| - \frac{k}{2} \right| < 2\sqrt{k} \right] \leq 2e^{-\frac{2(2^2 k)}{k}} = 2e^{-8}$$

The result now follows by the union bound. $\qquad\square$

**Lemma 8.2.** *With probability $\geq 1 - \frac{\sqrt{2k}}{\sqrt{\pi}e^{8k}}$, for all $1 \leq j \leq k$ and $1 \leq i \leq 4$ it holds that $\left| \boldsymbol{w}_0^{(j)} \cdot \boldsymbol{x}_i \right| \leq \frac{\eta}{4}$ and $\left| \boldsymbol{u}_0^{(j)} \cdot \boldsymbol{x}_i \right| \leq \frac{\eta}{4}$.*

*Proof.* Let $Z$ be a random variable distributed as $\mathcal{N}(0, \sigma^2)$. Then by Proposition 2.1.2 in Vershynin (2017), we have

$$\mathbb{P}\left[ |Z| \geq t \right] \leq \frac{2\sigma}{\sqrt{2\pi}t} e^{-\frac{t^2}{2\sigma^2}}$$

Therefore, for all $1 \leq j \leq k$ and $1 \leq i \leq 4$,

$$\mathbb{P}\left[ \left| \boldsymbol{w}_0^{(j)} \cdot \boldsymbol{x}_i \right| \geq \frac{\eta}{4} \right] \leq \frac{1}{\sqrt{32\pi k}} e^{-8k}$$

and

$$\mathbb{P}\left[ \left| \boldsymbol{u}_0^{(j)} \cdot \boldsymbol{x}_i \right| \geq \frac{\eta}{4} \right] \leq \frac{1}{\sqrt{32\pi k}} e^{-8k}$$

The result follows by applying a union bound over all $2k$ weight vectors and the four points $\boldsymbol{x}_i$, $1 \leq i \leq 4$. $\qquad\square$

From now on we assume that the highly probable event in Lemma 8.2 holds.

**Lemma 8.3.** $N_{W_t}(\boldsymbol{x}^+) < 1$ and $-N_{W_t}(\boldsymbol{x}^-) < 1$ for $0 \leq t \leq 2$.

*Proof.* By Lemma 8.2 we have

$$N_{W_0}(\boldsymbol{x}^+) = \sum_{i=1}^{k} \left[ \max\left\{ \sigma\left( \boldsymbol{w}_0^{(i)} \cdot \boldsymbol{x}_1^+ \right), ..., \sigma\left( \boldsymbol{w}_0^{(i)} \cdot \boldsymbol{x}_d^+ \right) \right\} - \max\left\{ \sigma\left( \boldsymbol{u}_0^{(i)} \cdot \boldsymbol{x}_1^+ \right), ..., \sigma\left( \boldsymbol{u}_0^{(i)} \cdot \boldsymbol{x}_d^+ \right) \right\} \right]$$
$$\leq \frac{\eta k}{4} < \gamma$$

and similarly $-N_{W_0}(\boldsymbol{x}^-) < 1$. Therefore, by Eq. 10 and Eq. 11 we get:

1. For $i \in \{1,3\}$, $l \in \{2,4\}$, $j \in W_0^+(i) \cap W_0^-(l)$, it holds that $\boldsymbol{w}_1^{(j)} = \boldsymbol{w}_0^{(j)} - \eta\boldsymbol{x}_l + \eta\boldsymbol{x}_i$.

2. For $i \in \{2,4\}$ and $j \in W_0^+(i)$, it holds that $\boldsymbol{w}_1^{(j)} = \boldsymbol{w}_0^{(j)}$.

3. For $i \in \{1,3\}$, $l \in \{2,4\}$, $j \in U_0^+(i) \cap U_0^-(l)$, it holds that $\boldsymbol{u}_1^{(j)} = \boldsymbol{u}_0^{(j)} - \eta\boldsymbol{x}_i + \eta\boldsymbol{x}_l$.

4. For $i \in \{2,4\}$ and $j \in U_0^+(i)$, it holds that $\boldsymbol{u}_2^{(j)} = \boldsymbol{u}_0^{(j)}$.

Applying Lemma 8.2 again and using the fact that $\eta \leq \frac{1}{8k}$ we have $N_{W_1}(\boldsymbol{x}^+) < \gamma$ and $-N_{W_1}(\boldsymbol{x}^-) < 1$. Therefore we get,

1. For $i \in \{1, 3\}$, $l \in \{2, 4\}$, $j \in W_0^+(i) \cap W_0^-(l)$, it holds that $\boldsymbol{w}_2^{(j)} = \boldsymbol{w}_0^{(j)} + 2\eta \boldsymbol{x}_i$.

2. For $i \in \{2, 4\}$ and $j \in W_0^+(i)$, it holds that $\boldsymbol{w}_2^{(j)} = \boldsymbol{w}_0^{(j)}$.

3. For $i \in \{1, 3\}$, $l \in \{2, 4\}$, $j \in U_0^+(i) \cap U_0^-(l)$, it holds that $\boldsymbol{u}_2^{(j)} = \boldsymbol{u}_0^{(j)} - \eta \boldsymbol{x}_i + \eta \boldsymbol{x}_l$.

4. For $i \in \{2, 4\}$ and $j \in U_0^+(i)$, it holds that $\boldsymbol{u}_2^{(j)} = \boldsymbol{u}_0^{(j)}$.

As before, by Lemma 8.2 we have $N_{W_2}(\boldsymbol{x}^+) < \gamma$ and $-N_{W_2}(\boldsymbol{x}^-) < 1$. $\qquad \square$

### 8.6.2 Auxiliary Lemmas

**Lemma 8.4.** *For all $t \geq 1$ we have $W_t^+(i) = W_0^+(i)$ for $i \in \{1, 3\}$.*

*Proof.* We will first prove that $W_0^+(i) \subseteq W_t^+(i)$ for all $t \geq 1$. To prove this, we will show by induction on $t \geq 1$, that for all $j \in W_0^+(i) \cap W_0^+(l)$, where $l \in \{2, 4\}$ the following holds:

1. $j \in W_t^+(i)$.

2. $\boldsymbol{w}_t^{(j)} \cdot \boldsymbol{x}_l = \boldsymbol{w}_0^{(j)} \cdot \boldsymbol{x}_l - \eta$ or $\boldsymbol{w}_t^{(j)} \cdot \boldsymbol{x}_l = \boldsymbol{w}_t^{(0)} \cdot \boldsymbol{x}_l$.

3. $\boldsymbol{w}_t^{(j)} \cdot \boldsymbol{x}_i > \eta$.

The claim holds for $t = 1$ by the proof of Lemma 8.3. Assume it holds for $t = T$. By the induction hypothesis there exists an $l' \in \{2, 4\}$ such that $j \in W_T^+(i) \cap W_T^-(l')$. By Eq. 10 we have,

$$\boldsymbol{w}_{T+1}^{(j)} = \boldsymbol{w}_T^{(j)} + a\eta \boldsymbol{x}_i + b\eta \boldsymbol{x}_{l'} \tag{12}$$

where $a \in \{0, 1\}$ and $b \in \{-1, 0\}$.

If $\boldsymbol{w}_T^{(j)} \cdot \boldsymbol{x}_l = \boldsymbol{w}_0^{(j)} \cdot \boldsymbol{x}_l$ then $l' = l$ and either $\boldsymbol{w}_{T+1}^{(j)} \cdot \boldsymbol{x}_l = \boldsymbol{w}_0^{(j)} \cdot \boldsymbol{x}_l$ if $b = 0$ or $\boldsymbol{w}_{T+1}^{(j)} \cdot \boldsymbol{x}_l = \boldsymbol{w}_0^{(j)} \cdot \boldsymbol{x}_l - \eta$ if $b = -1$. Otherwise, assume that $\boldsymbol{w}_T^{(j)} \cdot \boldsymbol{x}_l = \boldsymbol{w}_0^{(j)} \cdot \boldsymbol{x}_l - \eta$. By Lemma 8.2 we have $0 < \boldsymbol{w}_0^{(j)} \cdot \boldsymbol{x}_l < \frac{\eta}{4}$. Therefore $-\eta < \boldsymbol{w}_T^{(j)} \cdot \boldsymbol{x}_l < 0$ and $l' \neq l$. It follows that either $\boldsymbol{w}_{T+1}^{(j)} \cdot \boldsymbol{x}_l = \boldsymbol{w}_0^{(j)} \cdot \boldsymbol{x}_l - \eta$ if $b = 0$ or $\boldsymbol{w}_{T+1}^{(j)} \cdot \boldsymbol{x}_l = \boldsymbol{w}_0^{(j)} \cdot \boldsymbol{x}_l$ if $b = -1$. In both cases, we have $\left| \boldsymbol{w}_{T+1}^{(j)} \cdot \boldsymbol{x}_l \right| < \eta$. Furthermore, by Eq. 12, $\boldsymbol{w}_{T+1}^{(j)} \cdot \boldsymbol{x}_i \geq \boldsymbol{w}_T^{(j)} \cdot \boldsymbol{x}_i > \eta$. Hence, $\arg\max_{1 \leq l \leq 4} \boldsymbol{w}_{T+1}^{(j)} \cdot \boldsymbol{x}_l = i$ which by definition implies that $j \in W_{T+1}^+(i)$. This concludes the proof by induction which shows that $W_0^+(i) \subseteq W_t^+(i)$ for all $t \geq 1$.

In order to prove the lemma, it suffices to show that $W_0^+(2) \cup W_0^+(4) \subseteq W_t^+(2) \cup W_t^+(4)$. This follows since $\bigcup_{i=1}^4 W_t^+(i) = \{1, 2, ..., k\}$. We will show by induction on $t \geq 1$, that for all $j \in W_0^+(2) \cup W_0^+(4)$, the following holds:

1. $j \in W_t^+(2) \cap W_t^+(4)$.

2. $\boldsymbol{w}_t^{(j)} = \boldsymbol{w}_0^{(j)} + m\boldsymbol{x}_2$ for $m \in \mathbb{Z}$.

The claim holds for $t = 1$ by the proof of Lemma 8.3. Assume it holds for $t = T$. By the induction hypothesis $j \in W_T^+(2) \cap W_T^+(4)$. Assume without loss of generality that $j \in W_T^+(2)$. This implies that $j \in W_T^-(2)$ as well. Therefore, by Eq. 10 we have

$$\boldsymbol{w}_{T+1}^{(j)} = \boldsymbol{w}_T^{(j)} + a\eta \boldsymbol{x}_2 + b\eta \boldsymbol{x}_2 \tag{13}$$

where $a \in \{0, 1\}$ and $b \in \{0, -1\}$. By the induction hypothesis, $\boldsymbol{w}_{T+1}^{(j)} = \boldsymbol{w}_0^{(j)} + m\boldsymbol{x}_2$ for $m \in \mathbb{Z}$. If $a = 1$ or $b = 0$ we have for $i \in \{1, 3\}$,

$$\boldsymbol{w}_{T+1}^{(j)} \cdot \boldsymbol{x}_2 \geq \boldsymbol{w}_T^{(j)} \cdot \boldsymbol{x}_2 > \boldsymbol{w}_T^{(j)} \cdot \boldsymbol{x}_i = \boldsymbol{w}_{T+1}^{(j)} \cdot \boldsymbol{x}_i$$

where the first inequality follows since $j \in W_T^+(2)$ and the second by Eq. 13. This implies that $j \in W_{T+1}^+(2) \cap W_{T+1}^+(4)$.

Otherwise, assume that $a = 0$ and $b = -1$. By Lemma 8.2 we have $\boldsymbol{w}_0^{(j)} \cdot \boldsymbol{x}_2 < \frac{\eta}{4}$. Since $j \in W_T^+(2)$, it follows by the induction hypothesis that $\boldsymbol{w}_T^{(j)} = \boldsymbol{w}_0^{(j)} + m\boldsymbol{x}_2$, where $m \in \mathbb{Z}$ and $m \geq 0$. To see this, note that if $m < 0$, then $\boldsymbol{w}_T^{(j)} \cdot \boldsymbol{x}_2 < 0$ and $j \notin W_T^+(2)$, which is a contradiction. Let $i \in \{1, 3\}$. If $m = 0$, then $\boldsymbol{w}_{T+1}^{(j)} = \boldsymbol{w}_0^{(j)} - \boldsymbol{x}_2$, $\boldsymbol{w}_{T+1}^{(j)} \cdot \boldsymbol{x}_4 > \frac{\eta}{2}$ and $\boldsymbol{w}_{T+1}^{(j)} \cdot \boldsymbol{x}_i = \boldsymbol{w}_0^{(j)} \cdot \boldsymbol{x}_i < \frac{\eta}{4}$ by Lemma 8.2. Therefore, $j \in W_{T+1}^+(4)$.

Otherwise, if $m > 0$, then $\boldsymbol{w}_{T+1}^{(j)} \cdot \boldsymbol{x}_2 \geq \boldsymbol{w}_0^{(j)} \cdot \boldsymbol{x}_2 > \boldsymbol{w}_0^{(j)} \cdot \boldsymbol{x}_i = \boldsymbol{w}_{T+1}^{(j)} \cdot \boldsymbol{x}_i$. Hence, $j \in W_{T+1}^+(2)$, which concludes the proof. $\quad\square$

**Lemma 8.5.** *For all $t \geq 0$ we have $U_0^+(2) \cup U_0^+(4) \subseteq U_t^+(2) \cup U_t^+(4)$.*

*Proof.* Let $j \in U_0^+(2) \cup U_0^+(4)$. It suffices to prove that $\boldsymbol{u}_t^{(j)} = \boldsymbol{u}_0^{(j)} + \alpha_t \eta \boldsymbol{x}_2$ for $\alpha_t \in \mathbb{Z}$. This follows since the inequalities $\left| \boldsymbol{u}_0^{(j)} \cdot \boldsymbol{x}_1 \right| < \left| \boldsymbol{u}_0^{(j)} \cdot \boldsymbol{x}_2 \right| \leq \frac{\eta}{4}$ imply that in this case $j \in U_t^+(2) \cup U_t^+(4)$. Assume by contradiction that there exist an iteration $t$ for which $\boldsymbol{u}_t^{(j)} = \boldsymbol{u}_0^{(j)} + \alpha_t \eta \boldsymbol{x}_2 + \beta_t \eta \boldsymbol{x}_i$ where $\beta_t \in \{-1, 1\}$, $\alpha_t \in \mathbb{Z}$, $i \in \{1, 3\}$ and $\boldsymbol{u}_{t-1}^{(j)} = \boldsymbol{u}_0^{(j)} + \alpha_{t-1} \eta \boldsymbol{x}_2$ where $\alpha_{t-1} \in \mathbb{Z}$. [9] Since the coefficient of $\boldsymbol{x}_i$ changed in iteration $t$, we have $j \in U_{t-1}^+(1) \cup U_{t-1}^+(3)$. However, this contradicts the claim above which shows that if $\boldsymbol{u}_{t-1}^{(j)} = \boldsymbol{u}_0^{(j)} + \alpha_{t-1} \eta \boldsymbol{x}_2$, then $j \in U_{t-1}^+(2) \cup U_{t-1}^+(4)$. $\quad\square$

**Lemma 8.6.** *Let $i \in \{1, 3\}$ and $l \in \{2, 4\}$. For all $t \geq 0$, if $j \in U_0^+(i) \cap U_0^-(l)$, then there exists $a_t \in \{0, -1\}$, $b_t \in \mathbb{N}$ such that $\boldsymbol{u}_t^{(j)} = \boldsymbol{u}_0^{(j)} + a_t \eta \boldsymbol{x}_i + b_t \eta \boldsymbol{x}_l$.*

*Proof.* First note that by Eq. 11 we generally have $\boldsymbol{u}_t^{(j)} = \boldsymbol{u}_0^{(j)} + \alpha \eta \boldsymbol{x}_i + \beta \eta \boldsymbol{x}_l$ where $\alpha, \beta \in \mathbb{Z}$. Since $\left| \boldsymbol{u}_0^{(j)} \cdot \boldsymbol{x}_1 \right| \leq \frac{\eta}{4}$, by the gradient update in Eq. 11 it holds that $a_t \in \{0, -1\}$. Indeed, $a_0 = 0$ and by the gradient update if $a_{t-1} = 0$ or $a_{t-1} = -1$ then $a_t \in \{-1, 0\}$.

Assume by contradiction that there exists an iteration $t > 0$ such that $b_t = -1$ and $b_{t-1} = 0$. Note that by Eq. 11 this can only occur if $j \in U_{t-1}^+(l)$. We have $\boldsymbol{u}_{t-1}^{(j)} = \boldsymbol{u}_0^{(j)} + a_{t-1} \eta \boldsymbol{x}_i$ where $a_{t-1} \in \{0, -1\}$. Observe that $\left| \boldsymbol{u}_{t-1}^{(j)} \cdot \boldsymbol{x}_i \right| \geq \left| \boldsymbol{u}_0^{(j)} \cdot \boldsymbol{x}_i \right|$ by the fact that $\left| \boldsymbol{u}_0^{(j)} \cdot \boldsymbol{x}_i \right| \leq \frac{\eta}{4}$. Since $\boldsymbol{u}_0^{(j)} \cdot \boldsymbol{x}_i > \boldsymbol{u}_0^{(j)} \cdot \boldsymbol{x}_l = \boldsymbol{u}_{t-1}^{(j)} \cdot \boldsymbol{x}_l$ we have $j \in U_{t-1}^+(1) \cup U_{t-1}^+(3)$, a contradiction. $\quad\square$

**Lemma 8.7.** *Let*

$$X_t^+ = \sum_{j \in W_t^+(1)} \left[ \max \left\{ \sigma \left( \boldsymbol{w}^{(i)} \cdot \boldsymbol{x}_1^+ \right), ..., \sigma \left( \boldsymbol{w}^{(i)} \cdot \boldsymbol{x}_d^+ \right) \right\} \right]$$

*and*

$$Y_t^+ = \sum_{j \in W_t^+(3)} \left[ \max \left\{ \sigma \left( \boldsymbol{w}^{(i)} \cdot \boldsymbol{x}_1^+ \right), ..., \sigma \left( \boldsymbol{w}^{(i)} \cdot \boldsymbol{x}_d^+ \right) \right\} \right]$$

*Then for all $t$, $\frac{X_t^+ - X_0^+}{\left| W_t^+(1) \right|} = \frac{Y_t^+ - Y_0^+}{\left| W_t^+(3) \right|}$.*

---

[9]Note that in each iteration $\beta_t$ changes by at most $\eta$.

*Proof.* We will prove the claim by induction on $t$. For $t = 0$ this clearly holds. Assume it holds for $t = T$. Let $j_1 \in W_T^+(1)$ and $j_2 \in W_T^+(3)$. By Eq. 10, the gradient updates of the corresponding weight vector are given as follows:

$$\boldsymbol{w}_{T+1}^{(j_1)} = \boldsymbol{w}_T^{(j_1)} + a\eta\boldsymbol{x}_1 + b_1\eta\boldsymbol{x}_2$$

and

$$\boldsymbol{w}_{T+1}^{(j_2)} = \boldsymbol{w}_T^{(j_2)} + a\eta\boldsymbol{x}_3 + b_2\eta\boldsymbol{x}_2$$

where $a \in \{0, 1\}$ and $b_1, b_2 \in \{-1, 0, 1\}$. By Lemma 8.4, $j_1 \in W_{T+1}^+(1)$ and $j_2 \in W_{T+1}^+(3)$. Therefore,

$$\max\left\{\sigma\left(\boldsymbol{w}_{T+1}^{(j_1)} \cdot \boldsymbol{x}_1^+\right), ..., \sigma\left(\boldsymbol{w}_{T+1}^{(j_1)} \cdot \boldsymbol{x}_d^+\right)\right\} = \max\left\{\sigma\left(\boldsymbol{w}_T^{(j_1)} \cdot \boldsymbol{x}_1^+\right), ..., \sigma\left(\boldsymbol{w}_T^{(j_1)} \cdot \boldsymbol{x}_d^+\right)\right\} + a\eta$$

and

$$\max\left\{\sigma\left(\boldsymbol{w}_{T+1}^{(j_2)} \cdot \boldsymbol{x}_1^+\right), ..., \sigma\left(\boldsymbol{w}_{T+1}^{(j_2)} \cdot \boldsymbol{x}_d^+\right)\right\} = \max\left\{\sigma\left(\boldsymbol{w}_T^{(j_2)} \cdot \boldsymbol{x}_1^+\right), ..., \sigma\left(\boldsymbol{w}_T^{(j_2)} \cdot \boldsymbol{x}_d^+\right)\right\} + a\eta$$

By Lemma 8.4 we have $\left|W_t^+(1)\right| = \left|W_0^+(1)\right|$ and $\left|W_t^+(3)\right| = \left|W_0^+(3)\right|$ for all $t$. It follows that

$$\frac{X_{T+1}^+ - X_0^+}{\left|W_{T+1}^+(1)\right|} = \frac{a\eta\left|W_0^+(1)\right| + X_T^+ - X_0^+}{\left|W_0^+(1)\right|}$$

$$= a\eta + \frac{Y_T^+ - Y_0^+}{\left|W_0^+(3)\right|}$$

$$= \frac{a\eta\left|W_0^+(3)\right| + Y_T^+ - Y_0^+}{\left|W_0^+(3)\right|}$$

$$= \frac{Y_{T+1}^+ - Y_0^+}{\left|W_{T+1}^+(3)\right|}$$

where the second equality follows by the induction hypothesis. This proves the claim. $\square$

### 8.6.3 BOUNDING $P_t^+$, $P_t^-$ AND $S_t^-$

**Lemma 8.8.** *The following holds*

1. $S_t^- \leq \left|W_t^+(1) \cup W_t^+(3)\right| \eta$ *for all* $t \geq 1$.

2. $P_t^+ \leq \left|U_t^+(1) \cup U_t^+(3)\right| \eta$ *for all* $t \geq 1$.

3. $P_t^- \leq \left|U_t^+(1) \cup U_t^+(3)\right| \eta$ *for all* $t \geq 1$.

*Proof.* In Lemma 8.4 we showed that for all $t \geq 0$ and $j \in W_t^+(1) \cup W_t^+(3)$ it holds that $\left|\boldsymbol{w}_t^{(j)} \cdot \boldsymbol{x}_2\right| \leq \eta$. This proves the first claim. The second claim follows similarly. Without loss of generality, let $j \in U_t^+(1)$. By Lemma 8.5 it holds that $U_{t'}^+(1) \subseteq U_0^+(1) \cup U_0^+(3)$ for all $t' \leq t$. Therefore, by Lemma 8.6 we have $\left|\boldsymbol{u}_t^{(j)}\boldsymbol{x}_1\right| < \eta$, from which the claim follows.

For the third claim, without loss of generality, assume by contradiction that for $j \in U_t^+(1)$ it holds that $\left|\boldsymbol{u}_t^{(j)} \cdot \boldsymbol{x}_2\right| > \eta$. Since $\left|\boldsymbol{u}_t^{(j)} \cdot \boldsymbol{x}_1\right| < \eta$ by Lemma 8.6, it follows that $j \in U_t^+(2) \cup U_t^+(4)$, a contradiction. Therefore, $\left|\boldsymbol{u}_t^{(j)} \cdot \boldsymbol{x}_2\right| \leq \eta$ for all $j \in U_t^+(1) \cup U_t^+(3)$, from which the claim follows. $\square$

### 8.6.4 DYNAMICS OF $S_t^+$

**Lemma 8.9.** *The following holds:*

1. *If* $N_{W_t}(\boldsymbol{x}^+) < \gamma$ *and* $-N_{W_t}(\boldsymbol{x}^-) < 1$, *then* $S_{t+1}^+ = S_t^+ + \eta\left|W_t^+(1) \cup W_t^+(3)\right|$.

2. *If $N_{W_t}(\boldsymbol{x}^+) \geq \gamma$ and $-N_{W_t}(\boldsymbol{x}^-) < 1$, then $S_{t+1}^+ = S_t^+$.*

3. *If $N_{W_t}(\boldsymbol{x}^+) < \gamma$ and $-N_{W_t}(\boldsymbol{x}^-) \geq 1$, then $S_{t+1}^+ = S_t^+ + \eta \left| W_t^+(1) \cup W_t^+(3) \right|$.*

*Proof.*    1. The equality follows since for each $i \in \{1,3\}$, $l \in \{2,4\}$ and $j \in W_t^+(i) \cap W_t^-(l)$ we have $\boldsymbol{w}_{t+1}^{(j)} = \boldsymbol{w}_t^{(j)} + \eta\boldsymbol{x}_i - \eta\boldsymbol{x}_l$ and $W_{t+1}^+(1) \cup W_{t+1}^+(3) = W_t^+(1) \cup W_t^+(3)$ by Lemma 8.4.

2. In this case for each $i \in \{1,3\}$, $l \in \{2,4\}$ and $j \in W_t^+(i) \cap W_t^-(l)$ we have $\boldsymbol{w}_{t+1}^{(j)} = \boldsymbol{w}_t^{(j)} - \eta\boldsymbol{x}_l$ and $W_{t+1}^+(1) \cup W_{t+1}^+(3) = W_t^+(1) \cup W_t^+(3)$ by Lemma 8.4.

3. This equality follows since for each $i \in \{1,3\}$, $l \in \{2,4\}$ and $j \in W_t^+(i) \cap W_t^-(l)$ we have $\boldsymbol{w}_{t+1}^{(j)} = \boldsymbol{w}_t^{(j)} + \eta\boldsymbol{x}_i$ and $W_{t+1}^+(1) \cup W_{t+1}^+(3) = W_t^+(1) \cup W_t^+(3)$ by Lemma 8.4.

$\square$

### 8.6.5    Upper Bounds on $N_{W_t}(\boldsymbol{x}^+)$, $-N_{W_t}(\boldsymbol{x}^-)$ and $S_t^+$

**Lemma 8.10.** *Assume that $N_{W_t}(\boldsymbol{x}^+) \geq \gamma$ and $-N_{W_t}(\boldsymbol{x}^-) < 1$ for $T \leq t < T + b$ where $b \geq 2$. Then $N_{W_{T+b}}(\boldsymbol{x}^+) \leq N_{W_T}(\boldsymbol{x}^+) - (b-1)c_\eta + \eta \left| W_0^+(2) \cup W_0^+(4) \right|$.*

*Proof.* Define $R_t^+ = Y_t^+ - Z_t^+$ where

$$Y_t^+ = \sum_{j \in W_t^+(2) \cup W_t^+(4)} \left[ \max \left\{ \sigma\left( \boldsymbol{w}^{(i)} \cdot \boldsymbol{x}_1^+ \right), ..., \sigma\left( \boldsymbol{w}^{(i)} \cdot \boldsymbol{x}_d^+ \right) \right\} \right]$$

and

$$Z_t^+ = \sum_{j \in U_t^+(2) \cup U_t^+(4)} \left[ \max \left\{ \sigma\left( \boldsymbol{u}^{(i)} \cdot \boldsymbol{x}_1^+ \right), ..., \sigma\left( \boldsymbol{u}^{(i)} \cdot \boldsymbol{x}_d^+ \right) \right\} \right]$$

Let $l \in \{2,4\}$, $t = T$ and $j \in U_{t+1}^+(l)$. Then, either $j \in U_t^+(2) \cup U_t^+(4)$ or $j \in U_t^+(1) \cup U_t^+(3)$. In the first case, $\boldsymbol{u}_{t+1}^{(j)} = \boldsymbol{u}_t^{(j)} + \eta\boldsymbol{x}_l$. Note that this implies that $U_t^+(2) \cup U_t^+(4) \subseteq U_{t+1}^+(2) \cup U_{t+1}^+(4)$ (since $\boldsymbol{x}_l$ will remain the maximal direction). Therefore,

$$\sum_{j \in \left( U_{t+1}^+(2) \cup U_{t+1}^+(4) \right) \bigcap \left( U_t^+(2) \cup U_t^+(4) \right)} \left[ \max \left\{ \sigma\left( \boldsymbol{u}_{t+1}^{(j)} \cdot \boldsymbol{x}_1^+ \right), ..., \sigma\left( \boldsymbol{u}_{t+1}^{(j)} \cdot \boldsymbol{x}_d^+ \right) \right\} \right]$$

$$- \sum_{j \in U_t^+(2) \cup U_t^+(4)} \left[ \max \left\{ \sigma\left( \boldsymbol{u}_t^{(j)} \cdot \boldsymbol{x}_1^+ \right), ..., \sigma\left( \boldsymbol{u}_{t+1}^{(j)} \cdot \boldsymbol{x}_d^+ \right) \right\} \right]$$

$$= \eta \left| \left( U_{t+1}^+(2) \cup U_{t+1}^+(4) \right) \bigcap \left( U_t^+(2) \cup U_t^+(4) \right) \right|$$

$$= \eta \left| U_t^+(2) \cup U_t^+(4) \right| \tag{14}$$

In the second case, where we have $j \in U_t^+(1) \cup U_t^+(3)$, it holds that $\boldsymbol{u}_{t+1}^{(j)} = \boldsymbol{u}_t^{(j)} + \eta\boldsymbol{x}_l$, $j \in U_t^-(l)$ and $\boldsymbol{u}_{t+1}^{(j)} \cdot \boldsymbol{x}_l > \eta$. Furthermore, by Lemma 8.6, $\boldsymbol{u}_t^{(j)} \cdot \boldsymbol{x}_i < \eta$ for $i \in \{1,3\}$. Note that by Lemma 8.6, any $j_1 \in U_t^+(1) \cup U_t^+(3)$ satisfies $j_1 \in U_{t+1}^+(2) \cup U_{t+1}^+(4)$. By all these observations, we have

$$\sum_{j \in \left( U_{t+1}^+(2) \cup U_{t+1}^+(4) \right) \bigcap \left( U_t^+(1) \cup U_t^+(3) \right)} \left[ \max \left\{ \sigma\left( \boldsymbol{u}_{t+1}^{(j)} \cdot \boldsymbol{x}_1^+ \right), ..., \sigma\left( \boldsymbol{u}_{t+1}^{(j)} \cdot \boldsymbol{x}_d^+ \right) \right\} \right]$$

$$- \sum_{j \in U_t^+(1) \cup U_t^+(3)} \left[ \max \left\{ \sigma\left( \boldsymbol{u}_t^{(j)} \cdot \boldsymbol{x}_1^+ \right), ..., \sigma\left( \boldsymbol{u}_{t+1}^{(j)} \cdot \boldsymbol{x}_d^+ \right) \right\} \right]$$

$$\geq 0 \tag{15}$$

By Eq. 14 and Eq. 15, it follows that, $Z_{t+1}^+ + P_{t+1}^+ \geq Z_{t+1}^+ \geq Z_t^+ + P_t^+ + \eta \left| U_t^+(2) \cup U_t^+(4) \right|$. By induction we have $Z_{t+b}^+ + P_{t+b}^+ \geq Z_t^+ + P_t^+ + \sum_{i=0}^{b-1} \eta \left| U_{t+i}^+(2) \cup U_{t+i}^+(4) \right|$. By Lemma 8.6 for any $1 \leq i \leq b-1$ we have $\left| U_{t+i}^+(2) \cup U_{t+i}^+(4) \right| = \{1, ..., k\}$. Therefore, $Z_{t+b}^+ + P_{t+b}^+ \geq Z_t^+ + P_t^+ + (b-1)c_\eta$.

Now, assume that $j \in W_T^+(l)$ for $l \in \{2, 4\}$. Then $\boldsymbol{w}_{T+1}^{(j)} = \boldsymbol{w}_T^{(j)} - \eta \boldsymbol{x}_l$. Thus either

$$\max \left\{ \sigma \left( \boldsymbol{w}_{T+1}^{(j)} \cdot \boldsymbol{x}_1^+ \right), ..., \sigma \left( \boldsymbol{w}_{T+1}^{(j)} \cdot \boldsymbol{x}_d^+ \right) \right\} - \max \left\{ \sigma \left( \boldsymbol{w}_T^{(j)} \cdot \boldsymbol{x}_1^+ \right), ..., \sigma \left( \boldsymbol{w}_T^{(j)} \cdot \boldsymbol{x}_d^+ \right) \right\} = -\eta$$

in the case that $j \in W_{T+1}^+(l)$, or

$$\max \left\{ \sigma \left( \boldsymbol{w}_{T+1}^{(j)} \cdot \boldsymbol{x}_1^+ \right), ..., \sigma \left( \boldsymbol{w}_{T+1}^{(j)} \cdot \boldsymbol{x}_d^+ \right) \right\} \leq \eta$$

if $j \notin W_{T+1}^+(l)$.

Applying these observations $b$ times, we see that $Y_{T+b}^+ - Y_T^+$ is at most $\eta \left| W_{T+b}^+(2) \cup W_{T+b}^+(4) \right| = \eta \left| W_0^+(2) \cup W_0^+(4) \right|$ where the equality follows by Lemma 8.4. By Lemma 8.9, we have $S_{T+b}^+ = S_T^+$.

Hence, we can conclude that

$$\begin{aligned} N_{W_{T+b}}(\boldsymbol{x}^+) - N_{W_T}(\boldsymbol{x}^+) &= S_{T+b}^+ + R_{T+b}^+ - P_{T+b}^+ - S_T^+ - R_T^+ + P_T^+ \\ &= Y_{T+b}^+ - Z_{T+b}^+ - P_{T+b}^+ - Y_T^+ + Z_T^+ + P_T^+ \\ &\leq -(b-1)c_\eta + \eta \left| W_0^+(2) \cup W_0^+(4) \right| \end{aligned}$$

$\square$

**Lemma 8.11.** *Assume that $N_{W_t}(\boldsymbol{x}^+) < \gamma$ and $-N_{W_t}(\boldsymbol{x}^-) \geq 1$ for $T \leq t < T + b$ where $b \geq 1$. Then $-N_{W_{T+b}}(\boldsymbol{x}^-) \leq -N_{W_T}(\boldsymbol{x}^-) - b\eta \left| W_0^+(2) \cup W_0^+(4) \right| + c_\eta$.*

*Proof.* Define

$$Y_t^- = \sum_{j \in W_t^+(2) \cup W_t^+(4)} \left[ \max \left\{ \sigma \left( \boldsymbol{w}^{(i)} \cdot \boldsymbol{x}_1^+ \right), ..., \sigma \left( \boldsymbol{w}^{(i)} \cdot \boldsymbol{x}_d^+ \right) \right\} \right]$$

and

$$Z_t^- = \sum_{j=1}^{k} \left[ \max \left\{ \sigma \left( \boldsymbol{u}^{(j)} \cdot \boldsymbol{x}_1^+ \right), ..., \sigma \left( \boldsymbol{u}^{(j)} \cdot \boldsymbol{x}_d^+ \right) \right\} \right]$$

First note that by Lemma 8.4 we have $W_{t+1}^+(2) \cup W_{t+1}^+(4) = W_t^+(2) \cup W_t^+(4)$. Next, for any $l \in \{2, 4\}$ and $j \in W_t^+(l)$ we have $\boldsymbol{w}_{t+1}^{(j)} = \boldsymbol{w}_t^{(j)} + \eta \boldsymbol{x}_l$. Therefore,

$$Y_{T+b}^- \geq Y_T^- + b\eta \left| W_T^+(2) \cup W_T^+(4) \right| = Y_T^- + b\eta \left| W_0^+(2) \cup W_0^+(4) \right|$$

where the second equality follows by Lemma 8.4.

Assume that $j \in U_T^+(l)$ for $l \in \{1, 3\}$. Then $\boldsymbol{u}_{T+1}^{(j)} = \boldsymbol{u}_T^{(j)} - \eta \boldsymbol{x}_l$ and

$$\max \left\{ \sigma \left( \boldsymbol{u}_{T+1}^{(j)} \cdot \boldsymbol{x}_1^- \right), ..., \sigma \left( \boldsymbol{u}_{T+1}^{(j)} \cdot \boldsymbol{x}_d^- \right) \right\} - \max \left\{ \sigma \left( \boldsymbol{u}_T^{(j)} \cdot \boldsymbol{x}_1^- \right), ..., \sigma \left( \boldsymbol{u}_T^{(j)} \cdot \boldsymbol{x}_d^- \right) \right\} = 0 \tag{16}$$

To see this, note that by Lemma 8.6 and Lemma 8.5 it holds that $\boldsymbol{u}_T^{(j)} = \boldsymbol{u}_0^{(j)} + a_T \eta \boldsymbol{x}_l$ where $a_T \in \{-1, 0\}$. Hence, $\boldsymbol{u}_{T+1}^{(j)} = \boldsymbol{u}_0^{(j)} + a_{T+1} \eta \boldsymbol{x}_l$ where $a_{T+1} \in \{-1, 0\}$. Since $\left| \boldsymbol{u}_0^{(j)} \cdot \boldsymbol{x}_2 \right| < \frac{\eta}{4}$ it follows that $\boldsymbol{u}_{T+1}^{(j)} \cdot \boldsymbol{x}_2 = \boldsymbol{u}_T^{(j)} \cdot \boldsymbol{x}_2 = \boldsymbol{u}_0^{(j)} \cdot \boldsymbol{x}_2$ and thus Eq. 16 holds.

Now assume that $j \in U_T^+(l)$ for $l \in \{2, 4\}$. Then

$$\max \left\{ \sigma \left( \boldsymbol{u}_{T+1}^{(j)} \cdot \boldsymbol{x}_1^- \right), ..., \sigma \left( \boldsymbol{u}_{T+1}^{(j)} \cdot \boldsymbol{x}_d^- \right) \right\} - \max \left\{ \sigma \left( \boldsymbol{u}_T^{(j)} \cdot \boldsymbol{x}_1^- \right), ..., \sigma \left( \boldsymbol{u}_T^{(j)} \cdot \boldsymbol{x}_d^- \right) \right\} = -\eta$$

if $l \in \{2,4\}$ and $j \in U_{T+1}^{+}(l)$, or

$$\max \left\{ \sigma \left( \boldsymbol{u}_{T+1}^{(j)} \cdot \boldsymbol{x}_1^{-} \right), ..., \sigma \left( \boldsymbol{u}_{T+1}^{(j)} \cdot \boldsymbol{x}_d^{-} \right) \right\} \leq \eta$$

if $l \in \{2,4\}$ and $j \notin U_{T+1}^{+}(l)$.

Applying these observations $b$ times, we see that $Z_{T+b}^{-} - Z_T^{-}$ is at most $\eta \left| U_{T+b}^{+}(2) \cup U_{T+b}^{+}(4) \right|$. Furthermore, for $j \in W_T^{+}(l)$, $l \in \{1,3\}$, it holds that $\boldsymbol{w}_{T+1}^{(j)} = \boldsymbol{w}_T^{(j)} + \eta \boldsymbol{x}_l$. Therefore

$$\max \left\{ \sigma \left( \boldsymbol{w}_{T+1}^{(j)} \cdot \boldsymbol{x}_1^{-} \right), ..., \sigma \left( \boldsymbol{w}_{T+1}^{(j)} \cdot \boldsymbol{x}_d^{-} \right) \right\} = \max \left\{ \sigma \left( \boldsymbol{w}_T^{(j)} \cdot \boldsymbol{x}_1^{-} \right), ..., \sigma \left( \boldsymbol{w}_T^{(j)} \cdot \boldsymbol{x}_d^{-} \right) \right\}$$

and since $W_{T+1}^{+}(1) \cup W_{T+1}^{+}(3) = W_T^{+}(1) \cup W_T^{+}(3)$ by Lemma 8.4, we get $S_{T+b}^{-} = S_T^{-}$. Hence, we can conclude that

$$
\begin{aligned}
-N_{W_{T+b}}(\boldsymbol{x}^{-}) + N_{W_T}(\boldsymbol{x}^{-}) &= -S_{T+b}^{-} - Y_{T+b}^{-} + Z_{T+b}^{-} + S_T^{-} + Y_T^{-} - Z_T^{-} \\
&\leq -b\eta \left| W_0^{+}(2) \cup W_0^{+}(4) \right| + \eta \left| U_{T+b}^{+}(2) \cup U_{T+b}^{+}(4) \right| \\
&\leq -b\eta \left| W_0^{+}(2) \cup W_0^{+}(4) \right| + c_\eta
\end{aligned}
$$

$\square$

**Lemma 8.12.** *For all $t$, $N_{W_t}(\boldsymbol{x}^{+}) \leq \gamma + 3c_\eta$, $-N_{W_t}(\boldsymbol{x}^{-}) \leq 1 + 3c_\eta$ and $S_t^{+} \leq \gamma + 1 + 8c_\eta$.*

*Proof.* The claim holds for $t = 0$. Consider an iteration $T$. If $N_{W_T}(\boldsymbol{x}^{+}) < \gamma$ then $N_{W_{T+1}}(\boldsymbol{x}^{+}) \leq N_{W_T}(\boldsymbol{x}^{+}) + 2\eta k \leq \gamma + 2c_\eta$. Now assume that $N_{W_t}(\boldsymbol{x}^{+}) \geq \gamma$ for $T \leq t \leq T+b$ and $N_{W_{T-1}}(\boldsymbol{x}^{+}) < \gamma$. By Lemma 8.10, it holds that $N_{W_{T+b}}(\boldsymbol{x}^{+}) \leq N_{W_T}(\boldsymbol{x}^{+}) + \eta k \leq N_{W_T}(\boldsymbol{x}^{+}) + c_\eta \leq \gamma + 3c_\eta$, where the last inequality follows from the previous observation. Hence, $N_{W_t}(\boldsymbol{x}^{+}) \leq \gamma + 3c_\eta$ for all $t$.

The proof of the second claim follows similarly. It holds that $-N_{W_{T+1}}(\boldsymbol{x}^{-}) < 1 + 2c_\eta$ if $-N_{W_T}(\boldsymbol{x}^{-}) < 1$. Otherwise if $-N_{W_t}(\boldsymbol{x}^{-}) \geq 1$ for $T \leq t \leq T+b$ and $-N_{W_{T-1}}(\boldsymbol{x}^{-}) < 1$ then $-N_{W_{T+b}}(\boldsymbol{x}^{-}) \leq 1 + 3c_\eta$ by Lemma 8.11.

The third claim holds by the following identities and bounds $N_{W_T}(\boldsymbol{x}^{+}) - N_{W_T}(\boldsymbol{x}^{-}) = S_T^{+} - P_T^{+} + P_T^{-} - S_T^{-}$, $P_T^{-} \geq 0$, $\left| P_T^{+} \right| \leq c_\eta$, $\left| S_T^{-} \right| \leq c_\eta$ and $N_{W_T}(\boldsymbol{x}^{+}) - N_{W_T}(\boldsymbol{x}^{-}) \leq \gamma + 1 + 6c_\eta$ by the previous claims. $\square$

### 8.6.6 OPTIMIZATION

We are now ready to prove a global optimality guarantee for gradient descent.

**Proposition 8.13.** *Let $k > 16$ and $\gamma \geq 1$. With probability at least $1 - \frac{\sqrt{2k}}{\sqrt{\pi} e^{8k}} - 4e^{-8}$, after $T = \frac{7(\gamma + 1 + 8c_\eta)}{\left( \frac{k}{2} - 2\sqrt{k} \right) \eta}$ iterations, gradient descent converges to a global minimum.*

*Proof.* First note that with probability at least $1 - \frac{\sqrt{2k}}{\sqrt{\pi} e^{8k}} - 4e^{-8}$ the claims of Lemma 8.1 and Lemma 8.2 hold. Now, if gradient descent has not reached a global minimum at iteration $t$ then either $N_{W_t}(\boldsymbol{x}^{+}) < \gamma$ or $-N_{W_t}(\boldsymbol{x}^{-}) < 1$. If $-N_{W_t}(\boldsymbol{x}^{+}) < \gamma$ then by Lemma 8.9 it holds that

$$S_{t+1}^{+} \geq S_t^{+} + \eta \left| W_0^{+}(1) \cup W_0^{+}(3) \right| \geq S_t^{+} + \left( \frac{k}{2} - 2\sqrt{k} \right) \eta \tag{17}$$

where the last inequality follows by Lemma 8.1.

If $N_{W_t}(\boldsymbol{x}^{+}) \geq \gamma$ and $-N_{W_t}(\boldsymbol{x}^{-}) < 1$ we have $S_{t+1}^{+} = S_t^{+}$ by Lemma 8.9. However, by Lemma 8.10, it follows that after 5 consecutive iterations $t < t' < t+6$ in which $N_{W_{t'}}(\boldsymbol{x}^{+}) \geq \gamma$ and $-N_{W_{t'}}(\boldsymbol{x}^{-}) < 1$, we have $N_{W_{t+6}}(\boldsymbol{x}^{+}) < \gamma$. To see this, first note that for all $t$, $N_{W_t}(\boldsymbol{x}^{+}) \leq \gamma + 3c_\eta$ by Lemma 8.12. Then, by Lemma 8.10 we have

$$
\begin{aligned}
N_{W_{t+6}}(\boldsymbol{x}^{+}) &\leq N_{W_t}(\boldsymbol{x}^{+}) - 5c_\eta + \eta \left| W_0^{+}(2) \cup W_0^{+}(4) \right| \\
&\leq \gamma + 3c_\eta - 5c_\eta + c_\eta \\
&< \gamma
\end{aligned}
$$

where the second inequality follows by Lemma 8.1 and the last inequality by the assumption on $k$.

Assume by contradiction that GD has not converged to a global minimum after $T = \frac{7(\gamma+1+8c_\eta)}{(\frac{k}{2}-2\sqrt{k})\eta}$ iterations. Then, by the above observations, and the fact that $S_0^+ > 0$ with probability 1, we have

$$S_T^+ \geq S_0^+ + \left(\frac{k}{2} - 2\sqrt{k}\right)\eta\frac{T}{7}$$
$$> \gamma + 1 + 8c_\eta$$

However, this contradicts Lemma 8.12. $\square$

### 8.6.7 GENERALIZATION ON POSITIVE CLASS

We will first need the following three lemmas.

**Lemma 8.14.** *With probability at least $1 - 4e^{-8}$, it holds that*

$$\left|\left|W_0^+(1)\right| - \frac{k}{4}\right| \leq 2\sqrt{k}$$

*and*

$$\left|\left|W_0^+(3)\right| - \frac{k}{4}\right| \leq 2\sqrt{k}$$

*Proof.* The proof is similar to the proof of Lemma 8.1. $\square$

**Lemma 8.15.** *Assume that gradient descent converged to a global minimum at iteration $T$. Then there exists an iteration $T_2 < T$ for which $S_t^+ \geq \gamma + 1 - 3c_\eta$ for all $t \geq T_2$ and for all $t < T_2$, $-N_{W_t}(\boldsymbol{x}^-) < 1$.*

*Proof.* Assume that for all $0 \leq t \leq T_1$ it holds that $N_{W_t}(\boldsymbol{x}^+) < \gamma$ and $-N_{W_t}(\boldsymbol{x}^-) < 1$. By continuing the calculation of Lemma 8.3 we have the following:

1. For $i \in \{1,3\}$, $l \in \{2,4\}$, $j \in W_0^+(i) \cap W_0^-(l)$, it holds that $\boldsymbol{w}_{T_1}^{(j)} = \boldsymbol{w}_0^{(j)} + T_1\eta\boldsymbol{x}_i - \frac{1}{2}(1 - (-1)^{T_1})\eta\boldsymbol{x}_l$ .

2. For $i \in \{2,4\}$ and $j \in W_0^+(i)$, it holds that $\boldsymbol{w}_{T_1}^{(j)} = \boldsymbol{w}_0^{(j)}$.

3. For $i \in \{1,3\}$, $l \in \{2,4\}$, $j \in U_0^+(i) \cap U_0^-(l)$, it holds that $\boldsymbol{u}_{T_1}^{(j)} = \boldsymbol{u}_0^{(j)} - \eta\boldsymbol{x}_i + \eta\boldsymbol{x}_l$.

4. For $i \in \{2,4\}$ and $j \in U_0^+(i)$, it holds that $\boldsymbol{u}_{T_1}^{(j)} = \boldsymbol{u}_0^{(j)}$.

Therefore, there exists an iteration $T_1$ such that $N_{W_{T_1}}(\boldsymbol{x}^+) \geq \gamma$ and $-N_{W_{T_1}}(\boldsymbol{x}^-) < 1$ and for all $t < T_1$, $N_{W_t}(\boldsymbol{x}^+) < \gamma$ and $-N_{W_t}(\boldsymbol{x}^-) < 1$. Let $T_2 \leq T$ be the first iteration such that $-N_{W_{T_2}}(\boldsymbol{x}^-) \geq 1$. We claim that for all $T_1 \leq t \leq T_2$ we have $N_{W_{T_1}}(\boldsymbol{x}^+) \geq \gamma - 2c_\eta$. It suffices to show that for all $T_1 \leq t < T_2$ the following holds:

1. If $N_{W_t}(\boldsymbol{x}^+) \geq \gamma$ then $N_{W_{t+1}}(\boldsymbol{x}^+) \geq \gamma - 2c_\eta$.

2. If $N_{W_t}(\boldsymbol{x}^+) < \gamma$ then $N_{W_{t+1}}(\boldsymbol{x}^+) \geq N_{W_t}(\boldsymbol{x}^+)$.

The first claim follows since at any iteration $N_{W_t}(\boldsymbol{x}^+)$ can decrease by at most $2\eta k = 2c_\eta$. For the second claim, let $t' < t$ be the latest iteration such that $N_{W_{t'}}(\boldsymbol{x}^+) \geq \gamma$. Then at iteration $t'$ it holds that $-N_{W_{t'}}(\boldsymbol{x}^-) < 1$ and $N_{W_{t'}}(\boldsymbol{x}^+) \geq \gamma$. Therefore, for all $i \in \{1,3\}$, $l \in \{2,4\}$ and $j \in U_0^+(i) \cap U_0^+(l)$ it holds that $\boldsymbol{u}_{t'+1}^{(j)} = \boldsymbol{u}_{t'}^{(j)} + \eta\boldsymbol{x}_l$. Hence, by Lemma 8.5 and Lemma 8.6 it holds that $U_{t'+1}^+(1) \cup U_{t'+1}^+(3) = \emptyset$. Therefore, by the gradient update in Eq. 11, for all $1 \leq j \leq k$, and all $t' < t'' \leq t$ we have $\boldsymbol{u}_{t''+1}^{(j)} = \boldsymbol{u}_{t''}^{(j)}$, which implies that $N_{W_{t''+1}}(\boldsymbol{x}^+) \geq N_{W_{t''}}(\boldsymbol{x}^+)$. For $t'' = t$ we get $N_{W_{t+1}}(\boldsymbol{x}^+) \geq N_{W_t}(\boldsymbol{x}^+)$.

The above argument shows that $N_{W_{T_2}}(\boldsymbol{x}^+) \geq \gamma - 2c_\eta$ and $-N_{W_{T_2}}(\boldsymbol{x}^-) \geq 1$. Since $N_{W_{T_2}}(\boldsymbol{x}^+) - N_{W_{T_2}}(\boldsymbol{x}^-) = S_{T_2}^+ - P_{T_2}^+ + P_{T_2}^- - S_{T_2}^-$, $P_{T_2}^-, S_{T_2}^- \geq 0$ and $\left| P_{T_2}^+ \right| \leq c_\eta$ it follows that $S_{T_2}^+ \geq \gamma + 1 - 3c_\eta$. Finally, by Lemma 8.9 we have $S_t^+ \geq \gamma + 1 - 3c_\eta$ for all $t \geq T_2$. $\square$

**Lemma 8.16.** *Let*

$$X_t^+ = \sum_{j \in W_t^+(2) \cup W_t^+(4)} \left[ \max \left\{ \sigma \left( \boldsymbol{w}^{(j)} \cdot \boldsymbol{x}_1^+ \right), ..., \sigma \left( \boldsymbol{w}^{(j)} \cdot \boldsymbol{x}_d^+ \right) \right\} \right]$$

*and*

$$Y_t^+ = \sum_{j \in U_t^+(2) \cup U_t^+(4)} \left[ \max \left\{ \sigma \left( \boldsymbol{u}^{(j)} \cdot \boldsymbol{x}_1^+ \right), ..., \sigma \left( \boldsymbol{u}^{(j)} \cdot \boldsymbol{x}_d^+ \right) \right\} \right]$$

*Assume that $k \geq 64$ and gradient descent converged to a global minimum at iteration $T$. Then, $X_T^+ \leq 34c_\eta$ and $Y_T^+ \leq 1 + 38c_\eta$.*

*Proof.* Notice that by the gradient update in Eq. 10 and Lemma 8.2, $X_t^+$ can be strictly larger than $\max \left\{ X_{t-1}^+, \eta \left| W_t^+(2) \cup W_t^+(4) \right| \right\}$ only if $N_{W_{t-1}}(\boldsymbol{x}^+) < \gamma$ and $-N_{W_{t-1}}(\boldsymbol{x}^-) \geq 1$. Furthermore, in this case $X_t^+ - X_{t-1}^+ = \eta \left| W_t^+(2) \cup W_t^+(4) \right|$. By Lemma 8.9, $S_t^+$ increases in this case by $\eta \left| W_t^+(1) \cup W_t^+(3) \right|$. We know by Lemma 8.15 that there exists $T_2 < T$ such that $S_{T_2}^+ \geq \gamma + 1 - 3c_\eta$ and that $N_{W_t}(\boldsymbol{x}^+) < \gamma$ and $-N_{W_t}(\boldsymbol{x}^-) \geq 1$ only for $t > T_2$. Since $S_t^+ \leq \gamma + 1 + 8c_\eta$ for all $t$ by Lemma 8.12, there can only be at most $\frac{11c_\eta}{\eta \left| W_T^+(1) \cup W_T^+(3) \right|}$ iterations in which $N_{W_t}(\boldsymbol{x}^+) < \gamma$ and $-N_{W_t}(\boldsymbol{x}^-) \geq 1$. It follows that

$$X_t^+ \leq \eta \left| W_T^+(2) \cup W_T^+(4) \right| + \frac{11c_\eta \eta \left| W_T^+(2) \cup W_T^+(4) \right|}{\eta \left| W_T^+(1) \cup W_T^+(3) \right|}$$

$$\leq c_\eta + 11c_\eta \frac{\left( \frac{k}{2} + 2\sqrt{k} \right)}{\left( \frac{k}{2} - 2\sqrt{k} \right)}$$

$$\leq 34c_\eta$$

where the second inequality follows by Lemma 8.1 and the third inequality by the assumption on $k$.

At convergence we have $N_{W_T}(\boldsymbol{x}^-) = S_T^- + X_T^+ - Y_T^+ - P_T^- \geq -1 - 3c_\eta$ by Lemma 8.12 (recall that $R_t^- = R_t^+ = X_t^+ - Y_t^+$). Furthermore, $P_T^- \geq 0$ and by Lemma 8.8 we have $S_T^- \leq c_\eta$. Therefore, we get $Y_T^+ \leq 1 + 38c_\eta$. $\square$

We are now ready to prove the main result of this section.

**Proposition 8.17.** *Define $\beta(\gamma) = \frac{\gamma - 40\frac{1}{4}c_\eta}{39c_\eta + 1}$. Assume that $\gamma \geq 2$ and $k \geq 64 \left( \frac{\beta(\gamma)+1}{\beta(\gamma)-1} \right)^2$. Then with probability at least $1 - \frac{\sqrt{2k}}{\sqrt{\pi}e^{8k}} - 8e^{-8}$, gradient descent converges to a global minimum which classifies all positive points correctly.*

*Proof.* With probability at least $1 - \frac{\sqrt{128k}}{\sqrt{\pi}e^{\frac{k}{2}}} - 8e^{-8}$ Proposition 8.13, and Lemma 8.14 hold. It suffices to show generalization on positive points. Assume that gradient descent converged to a global minimum at iteration $T$. Let $(\boldsymbol{z}, 1)$ be a positive point. Then there exists $\boldsymbol{z}_i \in \{(1, 1), (-1, -1)\}$. Assume without loss of generality that $\boldsymbol{z}_i = (-1, -1) = \boldsymbol{x}_3$. Define

$$X_t^+(i) = \sum_{j \in W_T^+(i)} \left[ \max \left\{ \sigma \left( \boldsymbol{w}^{(j)} \cdot \boldsymbol{x}_1^+ \right), ..., \sigma \left( \boldsymbol{w}^{(j)} \cdot \boldsymbol{x}_d^+ \right) \right\} \right]$$

$$Y_t^+(i) = \sum_{j \in U_T^+(i)} \left[ \max \left\{ \sigma \left( \boldsymbol{u}^{(j)} \cdot \boldsymbol{x}_1^+ \right), ..., \sigma \left( \boldsymbol{u}^{(j)} \cdot \boldsymbol{x}_d^+ \right) \right\} \right]$$

for $i \in [4]$.

Notice that

$$
\begin{aligned}
N_{W_T}(\boldsymbol{x}^+) &= X_T^+(1) + X_T^+(3) - P_T^+ + R_T^+ \\
&= X_T^+(1) + X_T^+(3) - P_T^+ + R_T^- \\
&= X_T^+(1) + X_T^+(3) - P_T^+ + N_{W_T}(\boldsymbol{x}^-) - S_T^- + P_T^-
\end{aligned}
$$

Since $N_{W_T}(\boldsymbol{x}^+) \geq \gamma$, $-N_{W_T}(\boldsymbol{x}^-) \geq 1$, $\left|P_T^-\right| \leq c_\eta$ by Lemma 8.8 and $P_T^+, S_T^- \geq 0$, we obtain

$$X_T^+(1) + X_T^+(3) \geq \gamma + 1 - c_\eta \tag{18}$$

Furthermore, by Lemma 8.7 we have

$$\frac{X_T^+(1) - X_0^+(1)}{\left|W_T^+(1)\right|} = \frac{X_T^+(3) - X_0^+(3)}{\left|W_T^+(3)\right|} \tag{19}$$

and by Lemma 8.14,

$$\frac{\frac{k}{4} - 2\sqrt{k}}{\frac{k}{4} + 2\sqrt{k}} \leq \frac{\left|W_T^+(1)\right|}{\left|W_T^+(3)\right|} \leq \frac{\frac{k}{4} + 2\sqrt{k}}{\frac{k}{4} - 2\sqrt{k}} \tag{20}$$

Let $\alpha(k) = \frac{\frac{k}{4} + 2\sqrt{k}}{\frac{k}{4} - 2\sqrt{k}}$. By Lemma 8.2 we have $\left|X_0^+(1)\right| \leq \frac{\eta k}{4} \leq \frac{c_\eta}{4}$. Combining this fact with Eq. 19 and Eq. 20 we get

$$X_T^+(1) \leq \alpha(k) X_T^+(3) + X_0^+(1) \leq \alpha(k) X_T^+(3) + \frac{c_\eta}{4}$$

which implies together with Eq. 18 that $X_T^+(3) \geq \frac{\gamma + 1 - \frac{5c_\eta}{4}}{1 + \alpha(k)}$. Therefore,

$$
\begin{aligned}
N_{W_T}(\boldsymbol{z}) &\geq X_T^+(3) - P_T^+ - Y_T^+(2) - Y_T^+(4) \\
&\geq \frac{\gamma + 1 - \frac{5c_\eta}{4}}{1 + \alpha(k)} - c_\eta - 1 - 3(8c_\eta) - 14c_\eta \\
&= \frac{\gamma + 1 - \frac{5c_\eta}{4}}{1 + \alpha(k)} - 39c_\eta - 1 > 0
\end{aligned} \tag{21}
$$

where the first inequality is true because

$$\sum_{j=1}^k \left[\max\left\{\sigma\left(\boldsymbol{u}^{(j)} \cdot \boldsymbol{z}_1\right), ..., \sigma\left(\boldsymbol{u}^{(j)} \cdot \boldsymbol{z}_d\right)\right\}\right] \leq \sum_{j=1}^k \left[\max\left\{\sigma\left(\boldsymbol{u}^{(j)} \cdot \boldsymbol{x}_1^+\right), ..., \sigma\left(\boldsymbol{u}^{(j)} \cdot \boldsymbol{x}_d^+\right)\right\}\right] \tag{22}$$

$$= P_T^+ + Y_T^+(2) + Y_T^+(4) \tag{23}$$

The second inequality in Eq. 21 follows since $P_T^+ \leq c_\eta$ and by appyling Lemma 8.16. Finally, the last inequality in Eq. 21 follows by the assumption on $k$. [10] Hence, $\boldsymbol{z}$ is classified correctly. $\qquad\square$

### 8.6.8 GENERALIZATION ON NEGATIVE CLASS

We will need the following lemmas.

**Lemma 8.18.** *With probability at least* $1 - 8e^{-8}$*, it holds that*

$$\left|\left|U_0^+(2)\right| - \frac{k}{4}\right| \leq 2\sqrt{k}$$

$$\left|\left|U_0^+(4)\right| - \frac{k}{4}\right| \leq 2\sqrt{k}$$

$$\left|\left|(U_0^+(1) \cup U_0^+(3)) \cap U_0^-(2)\right| - \frac{k}{4}\right| \leq 2\sqrt{k}$$

$$\left|\left|(U_0^+(1) \cup U_0^+(3)) \cap U_0^-(4)\right| - \frac{k}{4}\right| \leq 2\sqrt{k}$$

---

[10] The inequality $\frac{\gamma + 1 - \frac{5c_\eta}{4}}{1 + \alpha(k)} - 39c_\eta - 1 > 0$ is equivalent to $\alpha(k) < \beta(\gamma)$ which is equivalent to $k > 64\left(\frac{\beta(\gamma) + 1}{\beta(\gamma) - 1}\right)^2$.

*Proof.* The proof is similar to the proof of Lemma 8.1 and follows from the fact that

$$
\begin{aligned}
\mathbb{P}\left[j \in U_0^+(2)\right] &= \mathbb{P}\left[j \in U_0^+(4)\right] \\
&= \mathbb{P}\left[j \in \left(U_0^+(1) \cup U_0^+(3)\right) \cap U_0^-(2)\right] \\
&= \mathbb{P}\left[j \in \left(U_0^+(1) \cup U_0^+(3)\right) \cap U_0^-(4)\right] \\
&= \frac{1}{4}
\end{aligned}
$$

$\square$

**Lemma 8.19.** *Let*

$$
X_t^- = \sum_{j \in U_0^+(2)} \left[\max\left\{\sigma\left(\boldsymbol{u}_t^{(j)} \cdot \boldsymbol{x}_1^-\right), ..., \sigma\left(\boldsymbol{u}_t^{(j)} \cdot \boldsymbol{x}_d^-\right)\right\}\right]
$$

*and*

$$
Y_t^- = \sum_{j \in U_0^+(4)} \left[\max\left\{\sigma\left(\boldsymbol{u}_t^{(j)} \cdot \boldsymbol{x}_1^-\right), ..., \sigma\left(\boldsymbol{u}_t^{(j)} \cdot \boldsymbol{x}_d^-\right)\right\}\right]
$$

*Then for all $t$, there exists $X, Y \geq 0$ such that $|X| \leq \eta\left|U_0^+(2)\right|$, $|Y| \leq \eta\left|U_0^+(4)\right|$ and $\frac{X_t^- - X}{\left|U_0^+(2)\right|} = \frac{Y_t^- - Y}{\left|U_0^+(4)\right|}$.*

*Proof.* First, we will prove that for all $t$ there exists $a_t \in \mathbb{Z}$ such that for $j_1 \in U_0^-(2)$ and $j_2 \in U_0^-(4)$ it holds that $\boldsymbol{u}_t^{(j_1)} = \boldsymbol{u}_0^{(j_1)} + a_t \eta \boldsymbol{x}_2$ and $\boldsymbol{u}_t^{(j_2)} = \boldsymbol{u}_0^{(j_2)} - a_t \eta \boldsymbol{x}_2$. [11] We will prove this by induction on $t$.

For $t = 0$ this clearly holds. Assume it holds for an iteration $t$. Let $j_1 \in U_0^-(2)$ and $j_2 \in U_0^-(4)$. By the induction hypothesis, there exists $a_T \in \mathbb{Z}$ such that $\boldsymbol{u}_t^{(j_1)} = \boldsymbol{u}_0^{(j_1)} + a_t \eta \boldsymbol{x}_2$ and $\boldsymbol{u}_t^{(j_2)} = \boldsymbol{u}_0^{(j_2)} - a_t \eta \boldsymbol{x}_2$. Since for all $1 \leq j \leq k$ it holds that $\left|\boldsymbol{u}_0^{(j)} \cdot \boldsymbol{x}_2\right| < \frac{\eta}{4}$, it follows that either $U_0^-(2) \subseteq U_t^-(2)$ and $U_0^-(4) \subseteq U_t^-(4)$ or $U_0^-(2) \subseteq U_t^-(4)$ and $U_0^-(4) \subseteq U_t^-(2)$. In either case, by Eq. 11, we have the following update at iteration $t + 1$:

$$
\boldsymbol{u}_{t+1}^{(j_1)} = \boldsymbol{u}_t^{(j_1)} + a\eta \boldsymbol{x}_2
$$

and

$$
\boldsymbol{u}_{t+1}^{(j_2)} = \boldsymbol{u}_t^{(j_2)} - a\eta \boldsymbol{x}_2
$$

where $a \in \{-1, 0, 1\}$. Hence, $\boldsymbol{u}_{t+1}^{(j_1)} = \boldsymbol{u}_0^{(j_1)} + (a_t + a)\eta \boldsymbol{x}_2$ and $\boldsymbol{u}_t^{(j_2)} = \boldsymbol{u}_0^{(j_2)} - (a_t + a)\eta \boldsymbol{x}_2$. This concludes the proof by induction.

Now, consider an iteration $t$, $j_1 \in U_0^+(2)$, $j_2 \in U_0^+(4)$ and the integer $a_t$ defined above. If $a_t \geq 0$ then

$$
\max\left\{\sigma\left(\boldsymbol{u}_t^{(j_1)} \cdot \boldsymbol{x}_1^-\right), ..., \sigma\left(\boldsymbol{u}_t^{(j_1)} \cdot \boldsymbol{x}_d^-\right)\right\} - \max\left\{\sigma\left(\boldsymbol{u}_0^{(j_1)} \cdot \boldsymbol{x}_1^-\right), ..., \sigma\left(\boldsymbol{u}_0^{(j_1)} \cdot \boldsymbol{x}_d^-\right)\right\} = \eta a_t
$$

and

$$
\max\left\{\sigma\left(\boldsymbol{u}_t^{(j_2)} \cdot \boldsymbol{x}_1^-\right), ..., \sigma\left(\boldsymbol{u}_t^{(j_2)} \cdot \boldsymbol{x}_d^-\right)\right\} - \max\left\{\sigma\left(\boldsymbol{u}_0^{(j_2)} \cdot \boldsymbol{x}_1^-\right), ..., \sigma\left(\boldsymbol{u}_0^{(j_2)} \cdot \boldsymbol{x}_d^-\right)\right\} = \eta a_t
$$

Define $X = X_0^-$ and $Y = Y_0^-$ then $|X| \leq \eta\left|U_0^-(2)\right|$, $|Y| \leq \eta\left|U_0^-(4)\right|$ and

$$
\frac{X_t^- - X}{\left|U_0^-(2)\right|} = \frac{\left|U_0^-(2)\right| \eta a_t}{\left|U_0^-(2)\right|} = \eta a_t = \frac{\left|U_0^-(4)\right| \eta a_t}{\left|U_0^-(4)\right|} = \frac{Y_t^- - Y}{\left|U_0^-(4)\right|}
$$

which proves the claim in the case that $a_t \geq 0$.

If $a_t < 0$ it holds that

---

[11] Recall that by Lemma 8.5 we know that $U_0^+(2) \cup U_0^+(4) \subseteq U_t^+(2) \cup U_t^+(4)$.

$$\max\left\{\sigma\left(\boldsymbol{u}_t^{(j_1)}\cdot\boldsymbol{x}_1^-\right),...,\sigma\left(\boldsymbol{u}_t^{(j_1)}\cdot\boldsymbol{x}_d^-\right)\right\}-\max\left\{\sigma\left(\left(\boldsymbol{u}_0^{(j_1)}-\boldsymbol{x}_2\right)\cdot\boldsymbol{x}_1^-\right),...,\sigma\left(\left(\boldsymbol{u}_0^{(j_1)}-\boldsymbol{x}_2\right)\cdot\boldsymbol{x}_d^-\right)\right\}=\eta(-a_t-1)$$

and

$$\max\left\{\sigma\left(\boldsymbol{u}_t^{(j_2)}\cdot\boldsymbol{x}_1^-\right),...,\sigma\left(\boldsymbol{u}_t^{(j_2)}\cdot\boldsymbol{x}_d^-\right)\right\}-\max\left\{\sigma\left(\left(\boldsymbol{u}_0^{(j_2)}+\boldsymbol{x}_2\right)\cdot\boldsymbol{x}_1^-\right),...,\sigma\left(\left(\boldsymbol{u}_0^{(j_2)}+\boldsymbol{x}_2\right)\cdot\boldsymbol{x}_d^-\right)\right\}=\eta(-a_t-1)$$

Define

$$X=\sum_{j\in U_0^+(2)}\left[\max\left\{\sigma\left(\left(\boldsymbol{u}_0^{(j)}-\boldsymbol{x}_2\right)\cdot\boldsymbol{x}_1^-\right),...,\sigma\left(\left(\boldsymbol{u}_0^{(j)}-\boldsymbol{x}_2\right)\cdot\boldsymbol{x}_d^-\right)\right\}\right]$$

and

$$Y=\sum_{j\in U_0^+(4)}\left[\max\left\{\sigma\left(\left(\boldsymbol{u}_0^{(j)}+\boldsymbol{x}_2\right)\cdot\boldsymbol{x}_1^-\right),...,\sigma\left(\left(\boldsymbol{u}_0^{(j)}+\boldsymbol{x}_2\right)\cdot\boldsymbol{x}_d^-\right)\right\}\right]$$

Since for all $1\le j\le k$ it holds that $\left|\boldsymbol{u}_0^{(j)}\cdot\boldsymbol{x}_2\right|<\frac{\eta}{4}$, we have $|X|\le\eta\left|U_0^-(2)\right|$, $|Y|\le\eta\left|U_0^-(4)\right|$. Furthermore,

$$\frac{X_t^--X}{\left|U_0^-(2)\right|}=\frac{\left|U_0^-(2)\right|\eta(-a_t-1)}{\left|U_0^-(2)\right|}=\eta(-a_t-1)=\frac{\left|U_0^-(4)\right|\eta(-a_t-1)}{\left|U_0^-(4)\right|}=\frac{Y_t^--Y}{\left|U_0^-(4)\right|}$$

which concludes the proof. $\qquad\square$

**Lemma 8.20.** *Let*

$$X_t^-=\sum_{j\in\left(U_0^+(1)\cup U_0^+(3)\right)\cap U_0^-(2)}\left[\max\left\{\sigma\left(\boldsymbol{u}_t^{(j)}\cdot\boldsymbol{x}_1^-\right),...,\sigma\left(\boldsymbol{u}_t^{(j)}\cdot\boldsymbol{x}_d^-\right)\right\}\right]$$

*and*

$$Y_t^-=\sum_{j\in\left(U_0^+(1)\cup U_0^+(3)\right)\cap U_0^-(4)}\left[\max\left\{\sigma\left(\boldsymbol{u}_t^{(j)}\cdot\boldsymbol{x}_1^-\right),...,\sigma\left(\boldsymbol{u}_t^{(j)}\cdot\boldsymbol{x}_d^-\right)\right\}\right]$$

*Then for all $t$,* $\frac{X_t^--X_0^-}{\left|\left(U_0^+(1)\cup U_0^+(3)\right)\cap U_0^-(2)\right|}=\frac{Y_t^--Y_t^-}{\left|\left(U_0^+(1)\cup U_0^+(3)\right)\cap U_0^-(4)\right|}.$

*Proof.* We will first prove that for all $t$ there exists an integer $a_t\ge0$ such that for $j_1\in\left(U_0^+(1)\cup U_0^+(3)\right)\cap U_0^-(2)$ and $j_2\in\left(U_0^+(1)\cup U_0^+(3)\right)\cap U_0^-(4)$ it holds that $\boldsymbol{u}_t^{(j_1)}\cdot\boldsymbol{x}_2=\boldsymbol{u}_0^{(j_1)}\cdot\boldsymbol{x}_2+\eta a_t$ and $\boldsymbol{u}_t^{(j_2)}\cdot\boldsymbol{x}_4=\boldsymbol{u}_0^{(j_2)}\cdot\boldsymbol{x}_4+\eta a_t$. We will prove this by induction on $t$.

For $t=0$ this clearly holds. Assume it holds for an iteration $t$. Let $j_1\in\left(U_0^+(1)\cup U_0^+(3)\right)\cap U_0^-(2)$ and $j_2\in\left(U_0^+(1)\cup U_0^+(3)\right)\cap U_0^-(4)$. By the induction hypothesis, there exists an integer $a_t\ge0$ such that $\boldsymbol{u}_t^{(j_1)}\cdot\boldsymbol{x}_2=\boldsymbol{u}_0^{(j_1)}\cdot\boldsymbol{x}_2+\eta a_t$ and $\boldsymbol{u}_t^{(j_2)}\cdot\boldsymbol{x}_4=\boldsymbol{u}_0^{(j_2)}\cdot\boldsymbol{x}_4+\eta a_t$. Since for all $1\le j\le k$ it holds that $\left|\boldsymbol{u}_0^{(j)}\cdot\boldsymbol{x}_1\right|<\frac{\eta}{4}$, it follows that if $a_t\ge1$ we have the following update at iteration $T+1$:

$$\boldsymbol{u}_{t+1}^{(j_1)}=\boldsymbol{u}_t^{(j_1)}+a\eta\boldsymbol{x}_2$$

and

$$\boldsymbol{u}_{t+1}^{(j_2)}=\boldsymbol{u}_t^{(j_2)}+a\eta\boldsymbol{x}_4$$

where $a\in\{-1,0,1\}$. Hence, $\boldsymbol{u}_{t+1}^{(j_1)}\cdot\boldsymbol{x}_2=\boldsymbol{u}_0^{(j_1)}\cdot\boldsymbol{x}_2+\eta(a_t+a)$ and $\boldsymbol{u}_{t+1}^{(j_2)}\cdot\boldsymbol{x}_4=\boldsymbol{u}_0^{(j_2)}\cdot\boldsymbol{x}_4+\eta(a_t+a)$.

Otherwise, if $a_t=0$ then

$$\boldsymbol{u}_{t+1}^{(j_1)}=\boldsymbol{u}_t^{(j_1)}+a\eta\boldsymbol{x}_2+b_1\boldsymbol{x}_1$$

and

$$\boldsymbol{u}_{t+1}^{(j_2)}=\boldsymbol{u}_t^{(j_2)}+a\eta\boldsymbol{x}_4+b_2\boldsymbol{x}_1$$

such that $a \in \{0, 1\}$ and $b_1, b_2 \in \{-1, 0, 1\}$. Hence, $\boldsymbol{u}_{t+1}^{(j_1)} \cdot \boldsymbol{x}_2 = \boldsymbol{u}_0^{(j_1)} \cdot \boldsymbol{x}_2 + \eta(a_t + a)$ and $\boldsymbol{u}_{t+1}^{(j_2)} \cdot \boldsymbol{x}_4 = \boldsymbol{u}_0^{(j_2)} \cdot \boldsymbol{x}_4 + \eta(a_t + a)$. This concludes the proof by induction.

Now, consider an iteration $t$, $j_1 \in \left(U_0^+(1) \cup U_0^+(3)\right) \cap U_0^-(2)$ and $j_2 \in \left(U_0^+(1) \cup U_0^+(3)\right) \cap U_0^-(4)$ and the integer $a_t$ defined above. We have,

$$\max\left\{\sigma\left(\boldsymbol{u}_t^{(j_1)} \cdot \boldsymbol{x}_1^-\right), ..., \sigma\left(\boldsymbol{u}_t^{(j_1)} \cdot \boldsymbol{x}_d^-\right)\right\} - \max\left\{\sigma\left(\boldsymbol{u}_0^{(j_1)} \cdot \boldsymbol{x}_1^-\right), ..., \sigma\left(\boldsymbol{u}_0^{(j_1)} \cdot \boldsymbol{x}_d^-\right)\right\} = \eta a_t$$

and

$$\max\left\{\sigma\left(\boldsymbol{u}_t^{(j_2)} \cdot \boldsymbol{x}_1^-\right), ..., \sigma\left(\boldsymbol{u}_t^{(j_2)} \cdot \boldsymbol{x}_d^-\right)\right\} - \max\left\{\sigma\left(\boldsymbol{u}_0^{(j_2)} \cdot \boldsymbol{x}_1^-\right), ..., \sigma\left(\boldsymbol{u}_0^{(j_2)} \cdot \boldsymbol{x}_d^-\right)\right\} = \eta a_t$$

It follows that

$$\begin{aligned}
\frac{X_t^- - X_0^-}{\left|\left(U_0^+(1) \cup U_0^+(3)\right) \cap U_0^-(2)\right|} &= \frac{\left|\left(U_0^+(1) \cup U_0^+(3)\right) \cap U_0^-(2)\right| \eta a_t}{\left|\left(U_0^+(1) \cup U_0^+(3)\right) \cap U_0^-(2)\right|} \\
&= \eta a_t \\
&= \frac{\left|\left(U_0^+(1) \cup U_0^+(3)\right) \cap U_0^-(4)\right| \eta a_t}{\left|\left(U_0^+(1) \cup U_0^+(3)\right) \cap U_0^-(4)\right|} \\
&= \frac{Y_t^- - Y_0^-}{\left|\left(U_0^+(1) \cup U_0^+(3)\right) \cap U_0^-(4)\right|}
\end{aligned}$$

which concludes the proof. $\qquad\square$

We are now ready to prove the main result of this section.

**Proposition 8.21.** *Define* $\beta = \frac{1 - 36\frac{1}{4}c_\eta}{35 c_\eta}$. *Assume that* $k > 64 \left(\frac{\beta+1}{\beta-1}\right)^2$. *Then with probability at least* $1 - \frac{\sqrt{2k}}{\sqrt{\pi}e^{8k}} - 8e^{-8}$, *gradient descent converges to a global minimum which classifies all negative points correctly.*

*Proof.* With probability at least $1 - \frac{\sqrt{2k}}{\sqrt{\pi}e^{8k}} - 16e^{-8}$ Proposition 8.13 and Lemma 8.18 hold. It suffices to show generalization on negative points. Assume that gradient descent converged to a global minimum at iteration $T$. Let $(\boldsymbol{z}, -1)$ be a negative point. Assume without loss of generality that $\boldsymbol{z}_i = \boldsymbol{x}_2$ for all $1 \leq i \leq d$. Define the following sums for $l \in \{2, 4\}$,

$$X_t^- = \sum_{j \in W_t^+(2) \cup W_t^+(4)} \left[\max\left\{\sigma\left(\boldsymbol{w}^{(j)} \cdot \boldsymbol{x}_1^-\right), ..., \sigma\left(\boldsymbol{w}^{(j)} \cdot \boldsymbol{x}_d^-\right)\right\}\right]$$

$$Y_t^-(l) = \sum_{j \in U_0^+(l)} \left[\max\left\{\sigma\left(\boldsymbol{u}_t^{(j)} \cdot \boldsymbol{x}_1^-\right), ..., \sigma\left(\boldsymbol{u}_t^{(j)} \cdot \boldsymbol{x}_d^-\right)\right\}\right]$$

$$Z_t^-(l) = \sum_{j \in \left(U_0^+(1) \cup U_0^+(3)\right) \cap U_0^-(l)} \left[\max\left\{\sigma\left(\boldsymbol{u}^{(j)} \cdot \boldsymbol{x}_1^-\right), ..., \sigma\left(\boldsymbol{u}^{(j)} \cdot \boldsymbol{x}_d^-\right)\right\}\right]$$

First, we notice that

$$N_{W_T}(\boldsymbol{x}^-) = S_T^- + X_T^- - Y_T^-(2) - Y_T^-(4) - Z_T^-(2) - Z_T^-(4)$$

$$X_T^-, S_T^- \geq 0$$

and

$$N_{W_T}(\boldsymbol{x}^-) \leq -1$$

imply that

$$Y_T^-(2) + Y_T^-(4) + Z_T^-(2) + Z_T^-(4) \geq 1 \tag{24}$$

We note that by the analysis in Lemma 8.18, it holds that for any $t$, $j_1 \in U_0^+(2)$ and $j_2 \in U_0^+(4)$, either $j_1 \in U_t^+(2)$ and $j_2 \in U_t^+(4)$, or $j_1 \in U_t^+(4)$ and $j_2 \in U_t^+(2)$. We assume without loss of generality that $j_1 \in U_t^+(2)$ and $j_2 \in U_t^+(4)$. It follows that in this case $N_{W_T}(z) \leq S_T^- + X_T^- - Z_T^-(2) - Y_T^-(2)$. [12]Otherwise we would replace $Y_T^-(2)$ with $Y_T^-(4)$ and vice versa and continue with the same proof.

Let $\alpha(k) = \frac{\frac{k}{4} + 2\sqrt{k}}{\frac{k}{4} - 2\sqrt{k}}$. By Lemma 8.20 and Lemma 8.18

$$Z_T^-(4) \leq \alpha(k)Z_T^-(2) + Z_0^-(2) \leq \alpha(k)Z_T^-(2) + \frac{c_\eta}{4}$$

and by Lemma 8.19 and Lemma 8.18 there exists $Y \leq c_\eta$ such that:

$$Y_T^-(4) \leq \alpha(k)Y_T^-(2) + Y \leq \alpha(k)Y_T^-(2) + c_\eta$$

Plugging these inequalities in Eq. 24 we get:

$$\alpha(k)Z_T^-(2) + \frac{c_\eta}{4} + \alpha(k)Y_T^-(2) + c_\eta + Y_T^-(2) + Z_T^-(2) \geq 1$$

which implies that

$$Y_T^-(2) + Z_T^-(2) \geq \frac{1 - \frac{5c_\eta}{4}}{\alpha(k) + 1}$$

By Lemma 8.16 we have $X_T^- \leq 34c_\eta$. Hence, by using the inequality $S_T^- \leq c_\eta$ we conclude that

$$N_{W_T}(z) \leq S_T^- + X_T^- - Z_T^-(2) - Y_T^-(2) \leq 35c_\eta - \frac{1 - \frac{5c_\eta}{4}}{\alpha(k) + 1} < 0$$

where the last inequality holds for $k > 64\left(\frac{\beta+1}{\beta-1}\right)^2$. [13] Therefore, $z$ is classified correctly. □

### 8.6.9 FINISHING THE PROOF

First, for $k \geq 120$, with probability at least $1 - \frac{\sqrt{2k}}{\sqrt{\pi}e^{8k}} - 16e^{-8}$, Proposition 8.13, Lemma 8.14 and Lemma 8.18 hold. Also, for the bound on $T$, note that in this case $\frac{28(\gamma+1+8c_\eta)}{c_\eta} \geq \frac{7(\gamma+1+8c_\eta)}{\left(\frac{k}{2}-2\sqrt{k}\right)\eta}$. Define $\beta_1 = \frac{\gamma - 40\frac{1}{4}c_\eta}{39c_\eta + 1}$ and $\beta_2 = \frac{1 - 36\frac{1}{4}c_\eta}{35c_\eta}$ and let $\beta = \max\{\beta_1, \beta_2\}$. For $\gamma \geq 8$ and $c_\eta \leq \frac{1}{410}$ it holds that $64\left(\frac{\beta+1}{\beta-1}\right)^2 < 120$. By Proposition 8.17 and Proposition 8.21, it follows that for $k \geq 120$ gradient descent converges to a global minimum which classifies all points correctly.

We will now prove pattern detection results. In the case of over-paramterized networks, in Proposition 8.17 we proved that $X^+(1), X^+(3) \geq \frac{\gamma + 1 - \frac{5c_\eta}{4}}{1 + \alpha(k)}$. Since for $i \in \{1, 3\}$ it holds that $D_{x_i} \geq X^+(i)$, it follows that patterns $x_1$ and $x_3$ are detected. Similarly, in Proposition 8.21 our analysis implies that, without loss of generality, $Y_T^-(2) + Z_T^-(2), Y_T^-(4) + Z_T^-(4) \geq \frac{1 - \frac{5c_\eta}{4}}{\alpha(k)+1}$. Since, for $l \in \{2, 4\}$, $D_{x_l} \geq Y_T^-(l) + Z_T^-(l)$ (under the assumption that we assumed without loss of generality), it follows that patterns $x_2$ and $x_4$ are detected. The confidence of the detection is at least $\frac{1 - \frac{5c_\eta}{4}}{\alpha(k)+1}$.

### 8.7 PROOF OF THEOREM 5.3

1. We refer to Eq. 17 in the proof of Proposition 8.13. To show convergence and provide convergence rates of gradient descent, the proof uses Lemma 8.1. However, to only show

---

[12]The fact that we can omit the term $-Z_T^-(4)$ from the latter inequality follows from Lemma 8.6.

[13]It holds that $35c_\eta - \frac{1 - \frac{5c_\eta}{4}}{\alpha(k)+1} < 0$ if and only if $\alpha(k) < \beta$ which holds if and only if $k > 64\left(\frac{\beta+1}{\beta-1}\right)^2$.

convergence, it suffices to bound the probability that $W_0^+(1) \cup W_0^+(3) \neq \emptyset$ and that the initialization satisfies Lemma 8.2. Given that Lemma 8.2 holds (with probability at least $1 - \sqrt{\frac{8}{\pi}}e^{-32}$), then $W_0^+(1) \cup W_0^+(3) \neq \emptyset$ holds with probability $\frac{3}{4}$.

By the argument above, with probability at least $(p_+p_-)^m \left(1 - \sqrt{\frac{8}{\pi}}e^{-32}\right)\frac{3}{4}$ all training points are diverse, Lemma 8.2 holds with $k = 2$ and $W_0^+(1) \cup W_0^+(3) \neq \emptyset$ which implies that gradient descent converges to a global minimum. For the rest of the proof we will condition on the corresponding event. Let $T$ be the iteration in which gradient descent converges to a global minimum. Note that $T$ is a random variable. Denote the network at iteration $T$ by $N$. For all $\boldsymbol{z} \in \mathbb{R}^{2d}$ denote

$$N(\boldsymbol{z}) = \sum_{j=1}^{2} \left[ \max\left\{ \sigma\left(\boldsymbol{w}^{(j)} \cdot \boldsymbol{z}_1\right), ..., \sigma\left(\boldsymbol{w}^{(j)} \cdot \boldsymbol{z}_d\right)\right\} - \max\left\{ \sigma\left(\boldsymbol{u}^{(j)} \cdot \boldsymbol{z}_1\right), ..., \sigma\left(\boldsymbol{u}^{(j)} \cdot \boldsymbol{z}_d\right)\right\}\right]$$

Let $E$ denote the event for which at least one of the following holds:

(a) $W_T^+(1) = \emptyset$.
(b) $W_T^+(3) = \emptyset$.
(c) $\boldsymbol{u}^{(1)} \cdot \boldsymbol{x}_2 > 0$ and $\boldsymbol{u}^{(2)} \cdot \boldsymbol{x}_2 > 0$.
(d) $\boldsymbol{u}^{(1)} \cdot \boldsymbol{x}_4 > 0$ and $\boldsymbol{u}^{(2)} \cdot \boldsymbol{x}_4 > 0$.

Our proof will proceed as follows. We will first show that if $E$ occurs then gradient descent does not learn $f^*$, i.e., the network $N$ does not satisfy $\text{sign}\left(N(\boldsymbol{x})\right) = f^*(\boldsymbol{x})$ for all $\boldsymbol{x} \in \{\pm 1\}^{2d}$. Then, we will show that $\mathbb{P}\left[E\right] \geq \frac{11}{12}$. This will conclude the proof.

Assume that one of the first two items in the definition of the event $E$ occurs. Without loss of generality assume that $W_T^+(1) = \emptyset$ and recall that $\boldsymbol{x}^-$ denotes a negative vector which only contains the patterns $\boldsymbol{x}_2, \boldsymbol{x}_4$ and let $\boldsymbol{z}^+ \in \mathbb{R}^{2d}$ be a positive vector which only contains the patterns $\boldsymbol{x}_1, \boldsymbol{x}_2, \boldsymbol{x}_4$. By the assumption $W_T^+(1) = \emptyset$ and the fact that $\boldsymbol{x}_1 = -\boldsymbol{x}_3$ it follows that for all $j = 1, 2$,

$$\max\left\{ \sigma\left(\boldsymbol{w}^{(j)} \cdot \boldsymbol{z}_1^+\right), ..., \sigma\left(\boldsymbol{w}^{(j)} \cdot \boldsymbol{z}_d^+\right)\right\} = \max\left\{ \sigma\left(\boldsymbol{w}^{(j)} \cdot \boldsymbol{x}_1^-\right), ..., \sigma\left(\boldsymbol{w}^{(j)} \cdot \boldsymbol{x}_d^-\right)\right\}$$

Furthermore, since $\boldsymbol{z}^+$ contains more distinct patterns than $\boldsymbol{x}^-$, it follows that for all $j = 1, 2$,

$$\max\left\{ \sigma\left(\boldsymbol{u}^{(j)} \cdot \boldsymbol{z}_1^+\right), ..., \sigma\left(\boldsymbol{u}^{(j)} \cdot \boldsymbol{z}_d^+\right)\right\} \geq \max\left\{ \sigma\left(\boldsymbol{u}^{(j)} \cdot \boldsymbol{x}_1^-\right), ..., \sigma\left(\boldsymbol{u}^{(j)} \cdot \boldsymbol{x}_d^-\right)\right\}$$

Hence, $N(\boldsymbol{z}^+) \leq N(\boldsymbol{x}^-)$. Since at a global minimum $N(\boldsymbol{x}^-) \leq -1$, we have $N(\boldsymbol{z}^+) \leq -1$ and $\boldsymbol{z}_2$ is not classified correctly.

Now assume without loss of generality that the third item in the definition of $E$ occurs. Let $\boldsymbol{z}^-$ be the negative vector with all of its patterns equal to $\boldsymbol{x}_4$. It is clear that $N(\boldsymbol{z}^-) \geq 0$ and therefore $\boldsymbol{z}^-$ is not classified correctly. This concludes the first part of the proof. We will now proceed to show that $\mathbb{P}\left[E\right] \geq \frac{11}{12}$.

Denote by $A_i$ the event that item $i$ in the definition of $E$ occurs and for an event $A$ denote by $A^c$ its complement. Thus $E^c = \cap_{i=1}^4 A_i^c$ and $\mathbb{P}\left[E^c\right] = \mathbb{P}\left[A_3^c \cap A_4^c \mid A_1^c \cap A_2^c\right]\mathbb{P}\left[A_1^c \cap A_2^c\right]$. We will first calculate $\mathbb{P}\left[A_1^c \cap A_2^c\right]$. By Lemma 8.4, we know that for $i \in \{1, 3\}$, $W_0^+(i) = W_T^+(i)$. Therefore, it suffices to calculate the probabilty that $W_0^+(1) \neq \emptyset$ and $W_0^+(3) \neq \emptyset$, provided that $W_0^+(1) \cup W_0^+(3) \neq \emptyset$. Without conditioning on $W_0^+(1) \cup W_0^+(3) \neq \emptyset$, for each $1 \leq i \leq 4$ and $1 \leq j \leq 2$ the event that $j \in W_0^+(i)$ holds with probability $\frac{1}{4}$. Since the initializations of the filters are independent, we have $\mathbb{P}\left[A_1^c \cap A_2^c\right] = \frac{1}{6}$. [14]

We will show that $\mathbb{P}\left[A_3^c \cap A_4^c \mid A_1^c \cap A_2^c\right] = \frac{1}{2}$ by a symmetry argument. This will finish the proof of the theorem. For the proof, it will be more convenient to denote the matrix of weights at iteration $t$ as a tuple of 4 vectors, i.e., $W_t = \left(\boldsymbol{w}_0^{(1)}, \boldsymbol{w}_0^{(2)}, \boldsymbol{u}_0^{(1)}, \boldsymbol{u}_0^{(2)}\right)$. Consider two initializations $W_0^{(1)} = \left(\boldsymbol{w}_0^{(1)}, \boldsymbol{w}_0^{(2)}, \boldsymbol{u}_0^{(1)}, \boldsymbol{u}_0^{(2)}\right)$ and $W_0^{(2)} = \left(\boldsymbol{w}_0^{(1)}, \boldsymbol{w}_0^{(2)}, -\boldsymbol{u}_0^{(1)}, \boldsymbol{u}_0^{(2)}\right)$ and let $W_t^{(1)}$ and $W_t^{(2)}$ be the corresponding weight values at iteration $t$. We will prove the following lemma:

---

[14] Note that this holds after conditioning on the corresponding event of Lemma 8.2.

**Lemma 8.22.** *For all* $t \geq 0$, *if* $W_t^{(1)} = \left(\boldsymbol{w}_t^{(1)}, \boldsymbol{w}_t^{(2)}, \boldsymbol{u}_t^{(1)}, \boldsymbol{u}_t^{(2)}\right)$ *then* $W_t^{(2)} = \left(\boldsymbol{w}_t^{(1)}, \boldsymbol{w}_t^{(2)}, -\boldsymbol{u}_t^{(1)}, \boldsymbol{u}_t^{(2)}\right).$

*Proof.* We will show this by induction on $t$. [15]This holds by definition for $t = 0$. Assume it holds for an iteration $t$. Denote $W_{t+1}^{(2)} = (\boldsymbol{z}_1, \boldsymbol{z}_2, \boldsymbol{v}_1, \boldsymbol{v}_2)$. We need to show that $\boldsymbol{z}_1 = \boldsymbol{w}_{t+1}^{(1)}$, $\boldsymbol{z}_2 = \boldsymbol{w}_{t+1}^{(2)}$, $\boldsymbol{v}_1 = -\boldsymbol{u}_{t+1}^{(1)}$ and $\boldsymbol{v}_2 = \boldsymbol{u}_{t+1}^{(2)}$. By the induction hypothesis it holds that $N_{W_t^{(1)}}(\boldsymbol{x}^+) = N_{W_t^{(2)}}(\boldsymbol{x}^+)$ and $N_{W_t^{(1)}}(\boldsymbol{x}^-) = N_{W_t^{(2)}}(\boldsymbol{x}^-)$. This follows since for diverse points (either positive or negative), negating a neuron does not change the function value. Thus, according to Eq. 10 and Eq. 11 we have $\boldsymbol{z}_1 = \boldsymbol{w}_{t+1}^{(1)}$, $\boldsymbol{z}_2 = \boldsymbol{w}_{t+1}^{(2)}$ and $\boldsymbol{v}_2 = \boldsymbol{u}_{t+1}^{(2)}$. We are left to show that $\boldsymbol{v}_1 = -\boldsymbol{u}_{t+1}^{(1)}$. This follows from Eq. 11 and the following facts:

(a) $\boldsymbol{x}_3 = -\boldsymbol{x}_1$.
(b) $\boldsymbol{x}_2 = -\boldsymbol{x}_4$.
(c) $\arg\max_{1 \leq l \leq 4} \boldsymbol{u} \cdot \boldsymbol{x}_l = 1$ if and only if $\arg\max_{1 \leq l \leq 4} -\boldsymbol{u} \cdot \boldsymbol{x}_l = 3$.
(d) $\arg\max_{1 \leq l \leq 4} \boldsymbol{u} \cdot \boldsymbol{x}_l = 2$ if and only if $\arg\max_{1 \leq l \leq 4} -\boldsymbol{u} \cdot \boldsymbol{x}_l = 4$.
(e) $\arg\max_{l \in \{2,4\}} \boldsymbol{u} \cdot \boldsymbol{x}_l = 2$ if and only if $\arg\max_{l \in \{2,4\}} -\boldsymbol{u} \cdot \boldsymbol{x}_l = 4$.

To see this, we will illustrate this through one case, the other cases are similar. Assume, for example, that $\arg\max_{1 \leq l \leq 4} \boldsymbol{u}_t^{(1)} \cdot \boldsymbol{x}_l = 3$ and $\arg\max_{l \in \{2,4\}} \boldsymbol{u}_t^{(1)} \cdot \boldsymbol{x}_l = 2$ and assume without loss of generality that $N_{W_t^{(1)}}(\boldsymbol{x}^+) = N_{W_t^{(2)}}(\boldsymbol{x}^+) < \gamma$ and $N_{W_t^{(1)}}(\boldsymbol{x}^-) = N_{W_t^{(2)}}(\boldsymbol{x}^-) > -1$. Then, by Eq. 11, $\boldsymbol{u}_{t+1}^{(1)} = \boldsymbol{u}_t^{(1)} - \boldsymbol{x}_3 + \boldsymbol{x}_2$. By the induction hypothesis and the above facts it follows that $\boldsymbol{v}_1 = -\boldsymbol{u}_t^{(1)} - \boldsymbol{x}_1 + \boldsymbol{x}_4 = -\boldsymbol{u}_t^{(1)} + \boldsymbol{x}_3 - \boldsymbol{x}_2 = -\boldsymbol{u}_{t+1}^{(1)}$. This concludes the proof. $\square$

Consider an initialization of gradient descent where $\boldsymbol{w}_0^{(1)}$ and $\boldsymbol{w}_0^{(2)}$ are fixed and the event that we conditioned on in the beginning of the proof and $A_1^c \cap A_2^c$ hold. Define the set $B_1$ to be the set of all pair of vectors $(\boldsymbol{v}_1, \boldsymbol{v}_2)$ such that if $\boldsymbol{u}_0^{(1)} = \boldsymbol{v}_1$ and $\boldsymbol{u}_0^{(1)} = \boldsymbol{v}_2$ then at iteration $T$, $\boldsymbol{u}^{(1)} \cdot \boldsymbol{x}_2 > 0$ and $\boldsymbol{u}^{(2)} \cdot \boldsymbol{x}_2 > 0$. Note that this definition implicitly implies that this initialization satisfies the condition in Lemma 8.2 and leads to a global minimum. Similarly, let $B_2$ be the set of all pair of vectors $(\boldsymbol{v}_1, \boldsymbol{v}_2)$ such that if $\boldsymbol{u}_0^{(1)} = \boldsymbol{v}_1$ and $\boldsymbol{u}_0^{(1)} = \boldsymbol{v}_2$ then at iteration $T$, $\boldsymbol{u}^{(1)} \cdot \boldsymbol{x}_4 > 0$ and $\boldsymbol{u}^{(2)} \cdot \boldsymbol{x}_2 > 0$. First, if $(\boldsymbol{v}_1, \boldsymbol{v}_2) \in B_1$ then $(-\boldsymbol{v}_1, \boldsymbol{v}_2)$ satisfies the conditions of Lemma 8.2. Second, by Lemma 8.22, it follows that if $(\boldsymbol{v}_1, \boldsymbol{v}_2) \in B_1$ then initializing with $(-\boldsymbol{v}_1, \boldsymbol{v}_2)$, leads to the same values of $N_{W_t}(\boldsymbol{x}^+)$ and $N_{W_t}(\boldsymbol{x}^-)$ in all iterations $0 \leq t \leq T$. Therefore, initializing with $(-\boldsymbol{v}_1, \boldsymbol{v}_2)$ leads to a convergence to a global minimum with the same value of $T$ as the initialization with $(\boldsymbol{v}_1, \boldsymbol{v}_2)$. Furthermore, if $(\boldsymbol{v}_1, \boldsymbol{v}_2) \in B_1$, then by Lemma 8.22, initializing with $\boldsymbol{u}_0^{(1)} = -\boldsymbol{v}_1$ and $\boldsymbol{u}_0^{(1)} = \boldsymbol{v}_2$ results in $\boldsymbol{u}^{(1)} \cdot \boldsymbol{x}_2 < 0$ and $\boldsymbol{u}^{(2)} \cdot \boldsymbol{x}_2 > 0$. It follows that $(\boldsymbol{v}_1, \boldsymbol{v}_2) \in B_1$ if and only if $(-\boldsymbol{v}_1, \boldsymbol{v}_2) \in B_2$.

For $l_1, l_2 \in \{2, 4\}$ define $P_{l_1, l_2} = \mathbb{P}\left[\boldsymbol{u}^{(1)} \cdot \boldsymbol{x}_{l_1} > 0 \wedge \boldsymbol{u}^{(2)} \cdot \boldsymbol{x}_{l_2} > 0 \mid A_1^c \cap A_2^c, \boldsymbol{w}_0^{(1)}, \boldsymbol{w}_0^{(2)}\right]$ Then, by symmetry of the initialization and the latter arguments it follows that $P_{2,2} = P_{4,2}$. By similar arguments we can obtain the equalities $P_{2,2} = P_{4,2} = P_{4,4} = P_{2,4}$.

Since all of these four probabilities sum to 1, each is equal to $\frac{1}{4}$. [16]Taking expectations of these probabilities with respect to the values of $\boldsymbol{w}_0^{(1)}$ and $\boldsymbol{w}_0^{(2)}$ (given that Lemma 8.2 and $A_1^c \cap A_2^c$ hold) and using the law of total expectation, we conclude that

$$\mathbb{P}[A_3^c \cap A_4^c \mid A_1^c \cap A_2^c] = \mathbb{P}\left[\boldsymbol{u}^{(1)} \cdot \boldsymbol{x}_4 > 0 \wedge \boldsymbol{u}^{(2)} \cdot \boldsymbol{x}_2 > 0 \mid A_1^c \cap A_2^c\right]$$

$$+ \mathbb{P}\left[\boldsymbol{u}^{(1)} \cdot \boldsymbol{x}_2 > 0 \wedge \boldsymbol{u}^{(2)} \cdot \boldsymbol{x}_4 > 0 \mid A_1^c \cap A_2^c\right] = \frac{1}{2}$$

---

[15]Recall that we condition on the event corresponding to Lemma 8.2. By negating a weight vector we still satisfy the bounds in the lemma and therefore the claim that will follow will hold under this conditioning.
[16]Note that the probablity that $\boldsymbol{u}^{(i)} \cdot \boldsymbol{x}_j = 0$ is 0 for all possible $i$ and $j$.

Finally, we show results for detection of a pattern. To see this, we will show that if one of the four conditions of the event $E$ defined above is met, then for $c_d > 2c_\eta$, the network does not detect all patterns. If one of the last two conditions hold, then this is true even for $c_d \geq 0$. Now, assume without loss of generality that $W_T^+(1) = \emptyset$. In this case by Lemma 8.4 and Lemma 8.2, it follows that

$$D_{\boldsymbol{x_1}} = \sum_{j \in W_T^+(2) \cup W_T^+(4)} \left[ \sigma\left( \boldsymbol{w}_T^{(j)} \cdot \boldsymbol{x}_1 \right) \right] \leq 2c_\eta$$

and therefore, $\boldsymbol{x}_1$ cannot be detected with confidence greater than $2c_\eta$.

2. Let $\mathcal{Z}_1$ be the set of positive points which contain only the patterns $\boldsymbol{x}_1$, $\boldsymbol{x}_2$, $\boldsymbol{x}_4$, $\mathcal{Z}_2$ be the set of positive points which contain only the patterns $\boldsymbol{x}_3$, $\boldsymbol{x}_2$, $\boldsymbol{x}_4$. Let $\mathcal{Z}_3$ be the set which contains the negative point with all patterns equal to $\boldsymbol{x}_2$ and $\mathcal{Z}_4$ be the set which contains the negative point with all patterns equal to $\boldsymbol{x}_4$. By the proof of the previous section, if the event $E$ holds, then there exists $1 \leq i \leq 4$, such that gradient descent converges to a solution at iteration $T$ which errs on all of the points in $\mathcal{Z}_i$. Therefore, its test error will be at least $p^*$ (recall Eq. 3).

## 8.8 PROOF OF THEOREM 4.1

Let $\delta \geq 1 - p_+ p_- (1 - c - 16e^{-8})$. By Theorem 5.2, given 2 samples, one positive and one negative, with probability at least $1 - \delta \leq p_+ p_- (1 - c - 16e^{-8})$, gradient descent will converge to a global minimum that has 0 test error. Therefore, for all $\epsilon \geq 0$, $m(\epsilon, \delta) \leq 2$. On the other hand, by Theorem 5.3, if $m < \frac{2\log\left( \frac{48\delta}{33(1-c)} \right)}{\log(p_+ p_-)}$ then with probability greater than

$$(p_+ p_-)^{\frac{\log\left( \frac{48\delta}{33(1-c)} \right)}{\log(p_+ p_-)}} (1 - c)\frac{33}{48} = \delta$$

gradient descent converges to a global minimum with test error at least $p^*$. It follows that for $0 \leq \epsilon < p^*$, $m(\epsilon, \delta) \geq \frac{2\log\left( \frac{48\delta}{33(1-c)} \right)}{\log(p_+ p_-)}$.

# 9 XOR

In this section we assume that we are given a training set $S \subseteq \{\pm 1\}^2 \times \{\pm 1\}^2$ consisting of points $(\boldsymbol{x}_1, 1), (\boldsymbol{x}_2, -1), (\boldsymbol{x}_3, 1), (\boldsymbol{x}_4, -1)$, where $\boldsymbol{x}_1 = (1, 1)$, $\boldsymbol{x}_2 = (-1, 1)$, $\boldsymbol{x}_3 = (-1, -1)$ and $\boldsymbol{x}_4 = (1, -1)$. Our goal is to learn the XOR function with gradient descent.

Note that in the case of two dimensions, the convolutional network introduced in Section 3 reduces to the following two-layer fully connected network.

$$N_{W_t}(\boldsymbol{x}) = \sum_{i=1}^{k} \left[ \sigma\left( \boldsymbol{w}_t^{(i)} \cdot \boldsymbol{x} \right) - \sigma\left( \boldsymbol{u}_t^{(i)} \cdot \boldsymbol{x} \right) \right]$$

We consider running gradient descent with a constant learning rate $\eta \leq \frac{c_\eta}{k}$, $c_\eta \leq 1$ and IID gaussian initialization with mean 0 and standard deviation $\sigma_g = \frac{c_\eta}{16k^{3/2}}$. We assume that gradient descent minimizes the hinge loss

$$\ell(W) = \sum_{(\boldsymbol{x}, y) \in S} \max\{1 - y N_W(\boldsymbol{x}), 0\}$$

where optimization is only over the first layer. We will show that gradient descent converges to the global minimum in a constant number of iterations.

For each point $\boldsymbol{x}_i \in S$ define the following sets of neurons:

$$W_t^+(i) = \left\{ j \mid \boldsymbol{w}_t^{(j)} \cdot \boldsymbol{x}_i > 0 \right\}$$

$$W_t^-(i) = \left\{ j \mid \boldsymbol{w}_t^{(j)} \cdot \boldsymbol{x}_i < 0 \right\}$$

$$U_t^+(i) = \left\{ j \mid \boldsymbol{u}_t^{(j)} \cdot \boldsymbol{x}_i > 0 \right\}$$

$$U_t^+(i) = \left\{ j \mid \boldsymbol{u}_t^{(j)} \cdot \boldsymbol{x}_i < 0 \right\}$$

**Lemma 9.1.** *For $i \in \{1, 3\}$, if $j \in W_0^+(i)$ then $j \in W_t^+(i)$ for all $t > 0$ and if $j \in W_0^-(i)$ then $j \in W_t^-(i)$ for all $t > 0$. Similarly, for $i \in \{2, 4\}$, if $j \in U_t^+(i)$ then $j \in U_t^+(i)$ for all $t > 0$ and if $j \in W_t^-(i)$ then $j \in U_t^-(i)$ for all $t > 0$.*

*Proof.* Without loss of generality it suffices to prove the claim for $W_t^+(1)$. This follows by symmetry and the fact that $j \in W_t^+(1)$ if and only if $j \in W_t^-(3)$. The proof is by induction. Assume that $j \in W_{t,+}^i$. The gradient of $\boldsymbol{w}^{(i)}$ with respect to a point $(\boldsymbol{x}, y)$ is given by $\frac{\partial \ell_{(\boldsymbol{x},y)}}{\partial \boldsymbol{w}^{(i)}}(W) = -y\sigma'(\boldsymbol{w}^{(i)} \cdot \boldsymbol{x})\boldsymbol{x}\mathbb{1}_{yN_W(\boldsymbol{x})<1}$. Therefore, by the facts $\boldsymbol{w}_t^{(j)} \cdot \boldsymbol{x}_3 < 0$ and $\boldsymbol{x}_1 \cdot \boldsymbol{x}_2, \boldsymbol{x}_1 \cdot \boldsymbol{x}_4 = 0$ it follows that $\boldsymbol{w}_{t+1}^{(j)} \cdot \boldsymbol{x}_1 > 0$, which concludes the proof. □

**Lemma 9.2.** *With probability at least $1 - 8e^{-8}$, for all $1 \leq j \leq 4$*

$$\frac{k}{2} - 2\sqrt{k} \leq \left|W_0^+(j)\right|, \left|U_0^+(j)\right| \leq \frac{k}{2} + 2\sqrt{k}$$

*Proof.* Without loss of generality consider $\left|W_0^+(1)\right|$. Since the sign of a one dimensional Gaussian random variable is a Bernoulli random variable, we get by Hoeffding's inequality

$$\mathbb{P}\left(\left|\left|W_0^+(1)\right| - \frac{k}{2}\right| < 2\sqrt{k}\right) \leq 2e^{-\frac{2(2^2 k)}{k}} = 2e^{-8}$$

Since $\left|W_0^+(1)\right| + \left|W_0^+(3)\right| = k$ with probability 1, we get that if $\left|\left|W_0^+(1)\right| - \frac{k}{2}\right| < 2\sqrt{k}$ then $\left|\left|W_0^+(3)\right| - \frac{k}{2}\right| < 2\sqrt{k}$. The result now follows by symmetry and the union bound. □

For each point $\boldsymbol{x}_i$, define the following sums:

$$S_t^+(i) = \sum_{j \in W_t^+(i)} \sigma\left(\boldsymbol{w}_t^{(j)} \cdot \boldsymbol{x}_i\right)$$

$$S_t^-(i) = \sum_{j \in W_t^-(i)} \sigma\left(\boldsymbol{w}_t^{(j)} \cdot \boldsymbol{x}_i\right)$$

$$R_t^+(i) = \sum_{j \in U_t^+(i)} \sigma\left(\boldsymbol{u}_t^{(j)} \cdot \boldsymbol{x}_i\right)$$

$$R_t^-(i) = \sum_{j \in U_t^-(i)} \sigma\left(\boldsymbol{u}_t^{(j)} \cdot \boldsymbol{x}_i\right)$$

We will prove the following lemma regarding $S_t^+(1), S_t^-(1), R_t^+(1), R_t^-(1)$ for $i = 1$. By symmetry, analogous lemmas follow for $i \neq 1$.

**Lemma 9.3.** *The following holds with probability $\geq 1 - \frac{\sqrt{2k}}{\sqrt{\pi}e^{8k}}$:*

1. *For all $t \geq 0$, $R_t^+(1) + R_t^-(1) \leq k\eta$.*

2. *For all $t \geq 0$, $S_t^-(1) = 0$.*

3. *Let $t \geq 0$. If $-yN_{W_t}(\boldsymbol{x}_1) < 1$, then $S_{t+1}^+(1) \geq S_t^+(1) + \left|W_t^+(1)\right| \frac{\eta}{4}$. Otherwise, if $-yN_{W_t}(\boldsymbol{x}_1) \geq 1$ then $S_{t+1}^+(1) = S_t^+(1)$.*

*Proof.* 1. For $t = 0$ the claim holds by Lemma 8.2. Assume by contradiction that there exists $t > 0$, such that $R_t^+(1) + R_t^-(1) > k\eta$. It follows that, without loss of generality, there exists $j \in U_t^+(1)$ such that $\sigma\left(u_t^{(j)} \cdot x_1\right) > \eta$. In each iteration $u_t^{(j)} \cdot x_1$ can increase by at most $\eta$, therefore by Lemma 8.2 there exists $0 \le t' < t$ such that $0 < \sigma\left(u_{t'}^{(j)} \cdot x_1\right) \le \eta$ and $\sigma\left(u_{t'+1}^{(j)} \cdot x_1\right) > \eta$. However, in this case $u_{t'+1}^{(j)} = u_{t'}^{(j)} + \alpha x_1 + \beta x_2 + \gamma x_4$ where $\alpha \le 0$. Therefore, $u_{t'+1}^{(j)} \cdot x_1 < u_{t'}^{(j)} \cdot x_1 \le \eta$, a contradiction.

2. This is a direct consequence of Lemma 9.1.

3. By Lemma 9.1, if $j \in W_t^+(1)$ and $-yN_{W_t}(x_1) < 1$ then $w_{t+1}^{(j)} = w_t^{(j)} + \frac{\eta}{4}x_1 + \beta x_2 + \gamma x_4$, from which the first part of the claim follows. The second claim follows similarly.

$\square$

**Proposition 9.4.** *Assume that $k \ge 25$. With probability $\ge 1 - \frac{\sqrt{2k}}{\sqrt{\pi}e^{8k}} - 8e^{-8}$, for all $i$, if until iteration $T$ there were at least $l \ge \frac{k}{k-2\sqrt{k}} + 10$ iterations, in which $-yN_{W_t}(x_i) < 1$, then it holds that $-yN_{W_t}(x_i) \ge 1$ for all $t \ge T$.*

*Proof.* Without loss of generality assume that $i = 1$. By Lemma 9.3 and Lemma 8.2, with probability $\ge 1 - \frac{\sqrt{2k}}{\sqrt{\pi}e^{8k}} - 8e^{-8}$, if $-yN_{W_t}(x_1) < 1$ then $S_{t+1}^+(1) \ge S_t^+(1) + \left(\frac{k}{2} - 2\sqrt{k}\right)\eta$. Therefore, by Lemma 9.3, for all $t \ge T$

$$N_{W_t}(x) = S_t^+(1) + S_t^-(1) - R_t^+(1) + R_t^-(1)$$
$$\ge \left(\frac{k}{2} - 2\sqrt{k}\right)l\eta - k\eta$$
$$\ge 1$$

where the last ineqaulity follows by the assumption on $l$.

$\square$

**Theorem 9.5.** *Assume that $k \ge 25$. With probability $\ge 1 - \frac{\sqrt{2k}}{\sqrt{\pi}e^{8k}} - 8e^{-8}$, after at most $\frac{4k}{k-2\sqrt{k}} + 40$ iterations, gradient descent converges to a global minimum.*

*Proof.* Proposition 9.4 implies that there are at most $\frac{4k}{k-2\sqrt{k}} + 40$ iterations in which there exists $(x_i, y_i)$ such that $-y_iN_{W_t}(x_i) < 1$. After at most that many iterations, gradient descent converges to a global minimum.

$\square$

## 10 VC DIMENSION

As noted in Remark 3.1, the VC dimension of the model we consider is at most 15. To see this, we first define for any $z \in \{\pm 1\}^{2d}$ the set $P_z \subseteq \{\pm 1\}^2$ which contains all the distinct two dimensional binary patterns that $z$ has. For example, for a positive diverse point $z$ it holds that $P_z = \{\pm 1\}^2$. Now, for any points $z^{(1)}, z^{(2)} \in \{\pm 1\}^{2d}$ such that $P_{z^{(1)}} = P_{z^{(2)}}$ and for any filter $w \in \mathbb{R}^2$ it holds that $\max_j \sigma\left(w \cdot z_j^{(1)}\right) = \max_j \sigma\left(w \cdot z_j^{(2)}\right)$. Therefore, for any $W$, $N_W(z^{(1)}) = N_W(z^{(2)})$. Specifically, this implies that if both $z^{(1)}$ and $z^{(2)}$ are diverse then $N_W(z^{(1)}) = N_W(z^{(2)})$. Since there are 15 non-empty subsets of $\{\pm 1\}^2$, it follows that for any $k$ the network can shatter a set of at most 15 points, or equivalently, its VC dimension is at most 15. Despite these expressive power limitations, there is a generalization gap between small and large networks in this setting, as can be seen in Figure 1.

