# OpenReview forum: "Over-parameterization Improves Generalization in the XOR Detection Problem"
_ICLR.cc/2019/Conference_

### Official Review · AnonReviewer3 · 2018-10-29
**Highly Specialized Analysis for a Toy Problem**

**Rating:** 5
**Confidence:** 4

**Review:**

Summary of the paper:
This paper studies using a three-layer convolutional neural network for the XOR detection problem. The first layer consists of 2k 2 dimensional filters, the second layer is ReLU + max pooling and the third layer are k 1s and k (-1)s. This paper assumes the input data is generated from {-1,+1}^{2d} and a margin loss is used for training.
The main result is Theorem 4.1, which shows to achieve the same generalization error, defined as the difference between training and test error, the over-parameterized neural network needs significantly fewer samples than the non-over-parameterized one.
Theorem 5.2 and 5.3 further shows randomly initialized gradient descent can find a global minimum (I assume is 0?) for both small and large networks.


Major Comments:
1.  While this paper demonstrates some advantages of using over-parameterized neural networks, I have several concerns.
This is a very toy example, XORD problem with boolean cube input and non-overlapping filters. Furthermore, the entire analysis is highly tailored to this toy problem and it is very hard to see how it can be generalized to more practical settings like real-valued input.
2. The statement of Theorem 4.1 is not clear. The probabilities p_+ and p_- are induced by the distribution D. However, the statement is given p_+ and p_-, there exists one D satisfies certain properties.
3. In Theorem 5.1 and 5.2, the success probability decreases as the number of samples (m) increases.


Minor Comments:
1. The statement of Theorem 4.1 itself does not show the advantage of over-parameterization because optimization is not discussed. I suggest also adding discussion on the optimization to Sec.4 as well.
2. Page 5, last paragraph: (p1p-1)^m -> (p_+p_-)^m.
3. There are many typos in the references, e.g. cnn -> CNN, relu -> ReLU, xor -> XOR.

---

> ### Author Response · Authors · 2018-11-22
> **Response**
>
> We thank the reviewer for the thoughtful review. Below we respond to the comments raised by the reviewer.
>
> Major Comments (numbering corresponds to the numbering in the review):
>
> 1.    Please see the response to all reviewers above where we respond to this concern. Proving that overparameterization improves generalization in *any* learning task is very challenging. We show this for the first time in a novel setting which shares various characteristics with problems in practice.  We empirically show that our insights hold in a more general setting with non-binary inputs and 60 patterns in the data. We believe that our detailed analysis of gradient descent for XORD and concrete insights can guide results in more general settings.
> 2.    We revised the statement of the theorem. For the new statement we defined a new quantity p* in Equation 3 which is a property of the distribution. The theorem says that for any D with values p_+, p_- and p^* there is a data dependent generalization gap. The reason we previously chose a specific D is to get a precise range of values for epsilon. For different distributions the range of values of epsilon varies. Please see the details in Section 4.
> 3.     The success probability decreases with larger m because our analysis holds only when the training set consists of diverse points. Note that this suffices to prove a generalization gap for sufficiently large p_+ and p_-. We empirically show that a similar result holds for training sets that contain non-diverse points in Figure 1.
>
>
> Minor Comments:
>
> We revised the paper according to these comments.
> It seems that the ICLR bibliographic style makes all titles lower case in the references.

---

### Official Review · AnonReviewer2 · 2018-11-02
**Fixed labeling function?**

**Rating:** 5
**Confidence:** 4

**Review:**

The paper tries to offer an explanation about why over-parametrization can be helpful in neural networks; in particular, why over-parametrization can help having better generalization errors when we train the network with SGD and the activation functions are RELU.

The authors consider a particular setting where the labeling function is fixed (i.e., a certain XOR function). The SGD however does not use this information, and it is shown that SGD may converge to better global minimums when the network is over-parametrized.

The considered CNN is a basic one: only the weights of one layer is trained (others are fixed), and the only non-linearities are max-pooling and RELU (one can remove these two max-based operators with one appropriately defined max operator).

The simplicity of the CNN makes it unclear how much of the observed phenomenon is relevant to CNNs: Can the analysis made simpler by considering (appropriately-defined) linear classifiers instead of CNNs? Is there something inherently special about CNNs?

My main concern is, however, the combination of these two assumptions:
+ Labeling function is fixed
+ The distribution of data is of a certain form (i.e., Theorem 4.1 reads like: for every parameter p+ and p- there "exists" a distribution such that ...)

Isn't this too restrictive? For any two reasonable learning algorithms, there often exists a particular scenario (i.e., labeling function and distribution) that the first one could do better than the other.

On a minor note, the lower bound is proved for a certain range of parameters (similar to the upper bound). How do we know that these ranges are not specifically chosen so that they are "good" for the over-parametrized one and "bad" for the other?

--
I updated my score after reading other reviews and the authors' response.

---

> ### Author Response · Authors · 2018-11-22
> **Response**
>
> We thank the reviewer for the valuable feedback. Please see the response to all reviewers above. Below we address the concerns raised by the reviewer.
>
> Regarding the main concern, the labeling function is fixed but the theorem holds for many distributions. We have changed the statement of Theorem 4.1 to reflect this. For the new statement we defined a new quantity p* in Equation 3 which is a property of the distribution. The theorem says that for any D with values p_+, p_- and p^* there is a data dependent generalization gap. The reason we previously chose a specific D is to get a precise range of values for epsilon. For different distributions the range of values of epsilon varies. Please see the details in Section 4.
>
> Furthermore, as in the response to all reviewers above, we note here again that proving that overparameterization improves generalization in *any* learning task (i.e., fixed labeling function) is a major challenge.
>
> Regarding the CNN architecture, we note that although the network is simple from a practical perspective, analyzing such a network is very challenging. For example, we are not aware of any theoretical analysis of over-parameterized two-layer convolutional networks with max pooling. We chose a CNN because it empirically showed the same phenomenon that overparameterization improves generalization, it solves the XORD problem (i.e., learns the function) and we were able to analyze it. We note that the data is not linearly separable and therefore XORD cannot be solved with a linear classifier.
>
> Finally, we note that the range of parameters for theorem 5.3 is quite wide: it allows for *any* learning rate <= 1/82 and standard deviation <= 1/1900. These upper bounds should be higher in practice and are an artifact of the current analysis. We added experiments in Section 8.4 to show that the gap exists even for values outside these ranges.

---

> ### Author Response · Authors · 2018-12-03
> **Response to revision**
>
> We would appreciate it if the reviewer could elaborate on the revision. It is not clear what are the current concerns given our response.
>
> We have addressed the main concern regarding the fixed label function and certain distribution. Our result holds for many distributions. We have emphasized the significance and difficulty of analyzing the XORD and XOR problems.

---

### Official Review · AnonReviewer1 · 2018-11-06
**Interesting topic but study too focused on one particular case, without possibilities of generalization or new insight**

**Rating:** 4
**Confidence:** 4

**Review:**

The paper studies a particular task (the XOR detection problem) in a particular setup (see below), and proves mathematically that in that case, the training performs better when the number of features grows.

The task is the following one:
- consider a set of pairs of binary values (-1 or +1);
- detect whether at least one of these pairs is (+1, +1) or (-1, -1).

The design of the predictor is:
- for each pair, compute 2k features (of the form ReLu(linear combination of the values, without bias));
- compute the max over all pairs of these features (thus obtaining 2k values);
- return the k first values minus the k last ones.

The training set consists only of examples having the following property [named 'diversity']:
- if the example (which is a set of pairs) is negative (i.e. doesn't contain (+1,+1) nor (-1,-1)), then it contains both (-1,1) and (1,-1);
- if the example is positive, it contains all possible pairs.

The paper proves that, under this setup, training with a number of features k > 120 will perform better than with k = 2 only (while k = 2 is theoretically sufficient to solve the problem). While tackling an interesting problem (impact of over-parameterization), the proof is specific to this particular, unusual architecture, with a "max - max" over features independently computed for each pair that the example contains; it relies heavily on the fact that the input are binary, and that the number of possible input pairs is small (4), which implies that the features can take only 4 values. Note also that the probabilities in some theorems are not really probabilities of convergence/performance of the training algorithm per se (as one would expect in such PAC-looking bounds), but actually probabilities of the batch of examples to all satisfy some property (the diversity).

Thus it is difficult to get from this study any insight about the over-parameterization / training ability phenomenon, for more general tasks, datasets or architectures.
Though clearly an impressive amount of work has been done in this proof, I do not see how it can be generalized (there is no explanation in the paper in that regard either, while it would have been welcomed), and consequently be of interest for the vast majority of the ICLR community, which is why I call for rejection.

---

> ### Author Response · Authors · 2018-11-22
> **Response**
>
> We thank the reviewer for the detailed comments.
>
> We address the comment by the reviewer regarding significance and generalization of the result in the response to all reviewers above. We also summarize it here for convenience. Proving that overparameterization improves generalization in *any* learning task is very challenging. We show this for the first time in a novel setting which shares various characteristics with problems in practice.  We empirically show that our insights hold in a more general setting with non-binary inputs and 60 patterns in the data. We believe that our detailed analysis of gradient descent for XORD and concrete insights can guide results in more general settings. We also added a discussion of the generalization of the results to more advanced settings in the Conclusions section (which is essentially the last part of the comment to all reviewers above).
>
> Regarding the PAC looking bounds, note that our main result is a valid PAC generalization gap. Our convergence results, that hold under the assumption that the training set contains only diverse points, suffice for proving a generalization gap for distributions with sufficiently large p_+ and p_-. We also show empirically in Figure 1 that there is a generalization gap even for distributions with  p_+ and p_- that are not sufficiently large. Analyzing this case requires additional ideas and is left for future work.

---

### Author Response · Authors · 2018-11-22
**Response to all reviewers**

We thank the reviewers for their efforts. In this comment we include a response to all reviewers. We address the specific concerns of each reviewer in a separate comment under the corresponding review.

We would like to first emphasize again the main contribution of this paper. It is the first work in deep learning theory that explains the fact that larger models generalize better, by theoretically proving a gap in generalization error between smaller and larger models. Additional important contributions are a) Provide the first global optimality guarantee for learning a XOR function (in fact a more general case of XOR). b) Provide the first optimization guarantees for a network consisting of a ReLu layer, followed by a max-pooling layer.
Each of the three contributions above is an important challenge for deep learning theory, and in the paper we show all three. We believe this constitutes significant progress.

It seems that the main concerns of the reviewers are a) That we are considering a single simplified learning task and b) Whether the analysis can be generalized. Below we address these two concerns.

1. Study focused on simplified learning task:
We would first like to emphasize that proving that overparameterization *improves* generalization is extremely challenging due to two main reasons. First, to show a generalization *gap*, one needs to prove that large networks trained with gradient descent have better sample complexity than smaller ones. However, current generalization bounds that are based on complexity measures do not offer such guarantees (note that better generalization *upper* bounds for overparameterized networks do not prove this). Second, analyzing the dynamics of first-order methods on networks with ReLU activations is a major challenge. Indeed, there do not exist optimization guarantees even for very simple learning tasks such as the classic XOR problem in two dimensions (for which we provide guarantees in Section 9).
The above suggests that proving that overparameterization improves generalization in *any* learning task is highly non-trivial. Thus, it is natural to first try to understand this phenomenon in a simplified learning task. We believe that this approach can be fruitful for understanding neural networks in more complex tasks.
As a first step to tackle this problem, we devise XORD, a novel learning task which is an extension of the XOR problem, and provably show that overparameterization improves generalization. Although this is a simplified task, it shares salient features with problems in practice. Indeed, the goal is to solve a pattern recognition task with gradient descent trained on a 3-layer over-parameterized convolutional network with max pooling and a fully connected layer.

2. Significance and generalization of result:
Our result is the first to show a learning task where over-parameterization provably *improves* generalization for a neural network with ReLU activations. Our analysis reveals that in the XORD problem over-parameterized networks are biased towards global minima which detect more relevant patterns in the data. While we prove this only for the XORD problem and under the assumption that the training set contains diverse points, our experiments clearly show that a similar phenomenon occurs in other settings as well. We show that this is the case for XORD with non-diverse points (Figure 1) and in the more general OBD problem which contains 60 patterns in the data and is not restricted to binary inputs (Figure 2). Furthermore, our experiments on MNIST hint that this is the case in MNIST as well (Figure 3). By clustering the detected patterns of the large network we could achieve better accuracy with a small network. This suggests that the larger network detects more patterns with gradient descent even though its effective size is close to that of a small network. We believe that these insights and our detailed analysis can guide future work for showing similar results in other more complex tasks and provide better understanding of this phenomenon.

---

### Meta-Review · Area_Chair1 · 2018-12-11
**ICLR 2019 decision**

**Confidence:** 4
**Recommendation:** Reject

**Metareview:**

`This paper tackles the problem of learning with one hidden layer non-overlapping conv net for XOR detection problem. For this problem the paper shows that over parametrized models perform better, giving insights into why larger neural networks generalize better - an interesting question to study. However reviews opined that the setting considered in this paper is too specific to this XOR problem and the simplified network architecture,  and the techniques are not generalizable to other models. Generalizing these results to more complex architectures or other learning problems will make the paper more interesting.